# Antibody affinity versus dengue morphology influences neutralization

**Guntur Fibriansah**[1,2], **Elisa X. Y. Lim**[1,2], **Jan K. Marzinek**[3], **Thiam-Seng Ng**[1,2], **Joanne L. Tan**[1,2], **Roland G. Huber**[3], **Xin-Ni Lim**[1,2], **Valerie S. Y. Chew**[1,2], **Victor A. Kostyuchenko**[1,2], **Jian Shi**[2], **Ganesh S. Anand**[4], **Peter J. Bond**[3,4], **James E. Crowe, Jr.**[5,6]*, **Shee-Mei Lok**[1,2]*

1 Emerging Infectious Diseases, Duke–National University of Singapore Medical School, Singapore, Singapore, 2 Centre for BioImaging Sciences, National University of Singapore, Singapore, Singapore, 3 Bioinformatics Institute, A*STAR (Agency for Science, Technology and Research), Singapore, Singapore, 4 Department of Biological Sciences, National University of Singapore, Singapore, Singapore, 5 The Vanderbilt Vaccine Center, Vanderbilt University Medical Center, Nashville, Tennessee, United States of America, 6 Departments of Pediatrics and Pathology, Microbiology and Immunology, Vanderbilt University Medical Center, Nashville, Tennessee, United States of America

* james.crowe@vumc.org (JEC); sheemei.lok@duke-nus.edu.sg (SML)

**Data Availability Statement:** The cryo-EM maps of Class I and II of DENV1 strain WestPac 74-Fab 1C19 at 37˚C and DENV2 strain PVP94/07-Fab 1C19 complex at 37˚C or 40˚C were deposited in the Electron Microscopy Database under accession

## Abstract

Different strains within a dengue serotype (DENV1-4) can have smooth, or "bumpy" surface morphologies with different antigenic characteristics at average body temperature (37˚C). We determined the neutralizing properties of a serotype cross-reactive human monoclonal antibody (HMAb) 1C19 for strains with differing morphologies within the DENV1 and DENV2 serotypes. We mapped the 1C19 epitope to E protein domain II by hydrogen deuterium exchange mass spectrometry, cryoEM and molecular dynamics simulations, revealing that this epitope is likely partially hidden on the virus surface. We showed the antibody has high affinity for binding to recombinant DENV1 E proteins compared to those of DENV2, consistent with its strong neutralizing activities for all DENV1 strains tested regardless of their morphologies. This finding suggests that the antibody could out-compete E-to-E interaction for binding to its epitope. In contrast, for DENV2, HMAb 1C19 can only neutralize when the epitope becomes exposed on the bumpy-surfaced particle. Although HMAb 1C19 is not a suitable therapeutic candidate, this study with HMAb 1C19 shows the importance of choosing a high-affinity antibody that could neutralize diverse dengue virus morphologies for therapeutic purposes.

## Author summary

Dengue virus consists of four serotypes (DENV1-4) and there are different strains within a serotype. DENV can have smooth or bumpy surface morphologies at physiological body temperature of 37˚C, depending on the strain. We have determined the cryoEM structures of a cross-reactive neutralizing human monoclonal antibody (HMAb) 1C19 in complex with strains of DENV1 and DENV2 that form either smooth or bumpy surface morphologies. We have mapped the epitope of HMAb 1C19 to E protein domain II and

numbers EMD-30883, EMD-30884, EMD-30885 and EMD-30886, respectively. The modeled Class I and II DENV1 strain WestPac 74– Fab 1C19 complex structures were deposited in the Protein Data Bank under accession codes 7DWT and 7DWU, respectively.

**Funding:** This work was supported by National Research Foundation Singapore (NRF Investigatorship, NRF-NRFI2016-01) awarded to S-M.L. and National Institute of Allergy and Infectious Diseases grant U54 AI 057157 (the Region IV Southeast Regional Center of Excellence in Emerging Infections and Biodefense) awarded to J. E.C.; G.S.A., P.J.B and S.M.L thank National Research Foundation Singapore (NRF2017NRF-CRP001-027) for funding. The funders had no role in study design, data collection and analysis, decision to publish, or preparation of the manuscript.

**Competing interests:** I have read the journal's policy and the authors of this manuscript have the following competing interests: J.E.C. has served as a consultant for Lilly and Luna Biologics, is a member of the Scientific Advisory Boards of CompuVax and Meissa Vaccines and is Founder of IDBiologics. The Crowe laboratory at Vanderbilt University Medical Center has received sponsored research agreements from IDBiologics and AstraZeneca.

the epitope is likely partially hidden on the virus surface. We showed that the antibody has high affinity for binding to recombinant DENV1 E protein than to DENV2 E protein. This explains the strong neutralization activity for all DENV1 strains tested regardless of their morphologies at physiological temperature, whereas it can only neutralize DENV2 strain that exposes the epitope on the bumpy surface particles. These results suggest that high-affinity therapeutic antibodies could neutralize diverse dengue virus morphologies.

## Introduction

Dengue virus (DENV), a flavivirus, infects about 390 million people each year [1]. Although there is a licensed vaccine available in some countries, its efficacy is low [2,3], and there are no licensed antiviral drugs licensed to treat DENV infection [2–6]. DENV is spread through the bite of an infected *Aedes* mosquito [7]. Circulating DENV strains can be grouped into four serotypes (DENV1 to 4) [8,9]. Disease associated with DENV infection ranges in severity from asymptomatic, to mild febrile illness, to life-threatening severe dengue characterized by hemorrhagic fever or shock. The difficulty in making a good vaccine against dengue is in large part due to the possibility of "antibody-dependent enhancement" (ADE) [10]. Although infection with one serotype elicits lifelong protection against that particular serotype, in a second infection with another serotype, the antibodies already present inside the patient may not be able to neutralize the second serotype leading to enhancement of disease severity. Therefore, a safe and effective vaccine will have to incorporate viruses from all serotypes, and the host recipient must elicit equally high protective immunity against DENV1-4. Testing with each serotype individually can elicit protective immune responses in the host, but when the four serotypes are combined together, the protective immune response in the recipient may not be equal. Also, after vaccination, the natural gradual decrease in antibody levels with time may also potentially lead to ADE. Therefore, development of an effective dengue vaccine has been extremely challenging. Furthermore, our previous results [11,12] revealed that dengue virus can change its shape (smooth to bumpy surface or clubshape morphology) to escape from the immune system, and spontaneously undergo mutations that influence such shape changes, hence further complicating development of vaccines that may be prone to emergence of resistance. Therefore, it is important to investigate other possible therapies including antibody therapeutics. Here we study the effect of an antibody against different DENV morphologies.

   The outer protein layer of flavivirus particles consists of 180 copies of envelope (E) and membrane (M) protein heterodimers, which are anchored to a lipid bilayer membrane. The inside of the virus particle contains capsid proteins complexed with the single-stranded positive-sense RNA genome [13,14]. The E protein is the major target for neutralizing antibodies [15]. The E proteins on the virus surface are arranged in icosahedral symmetry with each of the asymmetric units containing three E proteins–designated molecules (mols) A, B and C. The 180 copies of E protein are presented as 90 head-to-tail homodimers, and three of the homodimers are positioned parallel to each other, forming a raft structure [13,14]. The E protein has three structural domains–designated DI, DII and DIII [16–19]. The hydrophobic fusion loop located on the tip of DII plays an important role in facilitating fusion of the virus lipid bilayer with the host endosomal membrane [20,21], whereas DIII binds to host receptors [22]. When virus is incubated at 37°C, the surface of some virus strains within a serotype can change from a smooth compact to a bumpy appearance [12,23]. On the surface of the bumpy particles, the inter-dimeric E protein interactions are loosened and have moved outwards. In this new conformation, the E proteins become more solvent accessible. This change can lead

to either a decrease or an increase in antibody binding, depending on whether the epitope for the antibody is disrupted or rendered more accessible, respectively [12,24].

Previous crystallography and cryo-electron microscopy (cryo-EM) complex structures of potent murine DENV neutralizing antibodies showed them to bind primarily to DIII [25–27]. In contrast, structures of potent neutralizing human monoclonal antibodies (HMAb) complexed with DENV showed the antibodies usually recognize quaternary structure-dependent epitopes on the compact virus structure conformation [28–31]. The mechanism of neutralization of such HMAbs appears to be mediated in part by locking the surface proteins in place, preventing dynamic changes necessary for infectivity. Another group of interesting antibodies includes serotype cross-reactive antibodies that bind to E protein DII, whose epitopes were identified by loss of antibody binding to various E protein mutants [32,33]. Of these antibodies, those that bind to the fusion loop typically mediate neutralization poorly, while others that bind to the *bc* loop in DII are more potent [33]. We previously showed that HMAb 1C19 neutralizes DENV serotypes 1, 2 and 3 with 50% maximum effective inhibitory concentration (IC$_{50}$) values in neutralization assays ranging from 0.01 to 0.05 µg/mL [33], and it binds to E protein monomers. The epitope of HMAb 1C19 was shown by site-directed mutagenesis to be affected by changes in the E protein *bc* loop [33].

Here, we used a combination of hydrogen-deuterium exchange mass spectrometry (HDXMS), molecular dynamics (MD) simulations, and cryo-EM to locate the 1C19 epitope on the E protein and also to determine the effect of antibody binding on virus particles. HDXMS and MD simulations using recombinant E protein of DENV1 showed that the footprint of the antibody likely covers half of DII near the fusion loop of an E protein protomer within a dimer, with the main interaction site at E protein residues 219 to 243. Combining these results with the cryoEM structure of the complex shows the epitope to be located in the middle portion of DII. However, the Fab may also interact transiently with DIII of the opposite protomer, as shown by HDXMS and MD studies of the recombinant E protein. This interaction with DIII is likely not critical, however, as the E protein dimers may separate when Fab is bound, as suggested by the cryoEM map of the Fab complexed with the whole virus. MD simulations also suggest the Fab:E protein interaction is highly dynamic. In cryoEM micrographs, we observed the binding of Fab 1C19 to two representative strains from each of the DENV1 and DENV2 serotypes. For each serotype, we chose to study one strain known to acquire a bumpy-surface morphology when incubated at 37°C, while the other strain remains smooth-surfaced. We compared the HMAb 1C19 neutralization potencies for these strains to analyze the correlation between morphology and neutralization potential. We observed that the neutralization mediated by HMAb 1C19 for DENV2 strains depends highly on whether the virus strain can undergo structural changes to bumpy-surfaced particles. In contrast, this HMAb can bind and neutralize all DENV1 virus morphologies efficiently. We also observed that the affinity of Fab 1C19 was higher for the DENV1 recombinant E protein than for that of DENV2. This work suggests that increasing the affinity of antibodies may help DENV neutralizing antibodies broaden their neutralization capabilities against diverse virus morphologies.

## Results

### Binding and neutralization profile of HMAb 1C19 to DENV1 or DENV2 strains at varying temperatures

For characterization of the binding and neutralizing properties of HMAb 1C19, two strains each of DENV1 (Western Pacific 1974—WestPac 74 and PVP159) and DENV2 (NGC and PVP94/07) were used. For each serotype, we chose a strain with a surface that becomes bumpy at 37°C and another that stays smooth. The cryo-EM micrographs of all un-complexed

DENV1 and DENV2 strains at 4˚C showed most particles with smooth surface, suggesting that there is very little contamination with immature virus (spikey particles). At 37˚C, ~50% of un-complexed DENV1 WestPac 74 strain particles became "bumpy" (Fig 1A), whereas PVP159 strain particles remained smooth and compact (Fig 1B). For the un-complexed DENV2 strains, the NGC strain has a bumpy-surfaced morphology at 37˚C [12] (Fig 1C), while the PVP94/07 remains smooth and becomes bumpy only when the temperature is increased to 40˚C (Fig 1D).

When equimolar ratios of Fab to E proteins on the virus surface were mixed at 4˚C, Fab 1C19 bound very few particles for all DENV1 and DENV2 strains, as shown by the presence of mostly smooth-surfaced particles in the cryoEM micrographs (Fig 1A–1D). At 37˚C, both DENV1 strains WestPac 74 and PVP159 were bound by Fab 1C19, as observed by the presence of particles with a spiky appearance (Figs 1A, 1B and S1A). In contrast, for DENV2 strains, only NGC with a bumpy-surfaced morphology allowed binding, observed by the larger diameter of particles compared to its un-complexed control particle preparation at the same temperature (Fig 1C). The smooth-surfaced PVP94/07 strain was observed to have ~60% of the virus particles bound by Fab 1C19 (Fig 1D). Of the Fab-bound DENV2 strain PVP94/07 particles, 2D class averages showed mixed populations of particles with varying degrees of Fab binding (S1B Fig), indicating low binding efficiency. At 40˚C, ~60% of the un-complexed DENV2 strain PVP94/07 particles acquired a bumpy morphology (Fig 1D), suggesting that high Fab 1C19 binding to DENV2 strain PVP94/07 could then be achieved. Indeed, the 2D class averages of the complex sample showed more homogeneously bound particles, with only a few unbound particles left compared with the same preparation at 37˚C (S1C Fig).

The inability of 1C19 to bind to all DENV1 and DENV2 strains at 4˚C suggests that the epitope is partially hidden at that temperature, and elevation of temperature may be required for movement of the E proteins, thereby increasing the exposure of the 1C19 epitope. Fab 1C19 bound to DENV1 strains regardless of their particle morphology at 37˚C, whereas increased efficiency of binding to DENV2 depended highly on increased epitope accessibility associated with the virus becoming bumpy.

We then conducted neutralization tests of HMAb 1C19 against these DENV1 and DENV2 strains with different morphologies (Table 1 and Fig 1E). At 37˚C, HMAb 1C19 exhibited high potency for neutralization of both DENV1 strains tested (Table 1)—PVP159 (neutralization $IC_{50}$ = 0.085 μg/mL) and strain Westpac 74 ($IC_{50}$ = 0.15 μg/mL), consistent with the cryoEM binding studies that showed high levels of antibody binding to both strains. This finding suggests that the antibody efficiently binds and neutralizes DENV1 viruses regardless of particle morphology at 37˚C. For DENV2, HMAb 1C19 failed to neutralize the smooth-surfaced DENV2 (strain PVP94/07) at 37˚C (Fig 1E and Table 1). Increasing the temperature to 40˚C increased the number of un-complexed bumpy-surfaced strain PVP94/07 particles in cryoEM, and the neutralization test at this temperature also showed the antibody was then able to neutralize the virus. For the bumpy-surfaced DENV2 strain (NGC), HMAb 1C19 potently neutralized the virus ($IC_{50}$ = 0.04 μg/mL). This finding suggests that neutralization of DENV2 by HMAb 1C19 depends highly on the ability of the virus to transition to a bumpy state in which the 1C19 epitope is likely more exposed.

We considered whether the differential binding patterns for DENV1 versus DENV2 particles we have observed were due to differences in affinity of binding of the antibody to DENV1 and DENV2 E proteins. We conducted biolayer interferometry (BLI) studies to determine the affinity of HMAb 1C19 for binding to recombinant DENV1 or DENV2 E proteins. The results showed that HMAb 1C19 has overall higher affinities (~5.4 to 26.7-fold higher) to DENV1 strains E proteins ($K_D$ = 0.712 and 0.844 nM for PVP159 and WestPac 74 strains, respectively) than to those of the DENV2 strains ($K_D$ = 19.015 and 4.549 nM for PVP94/07 and NGC

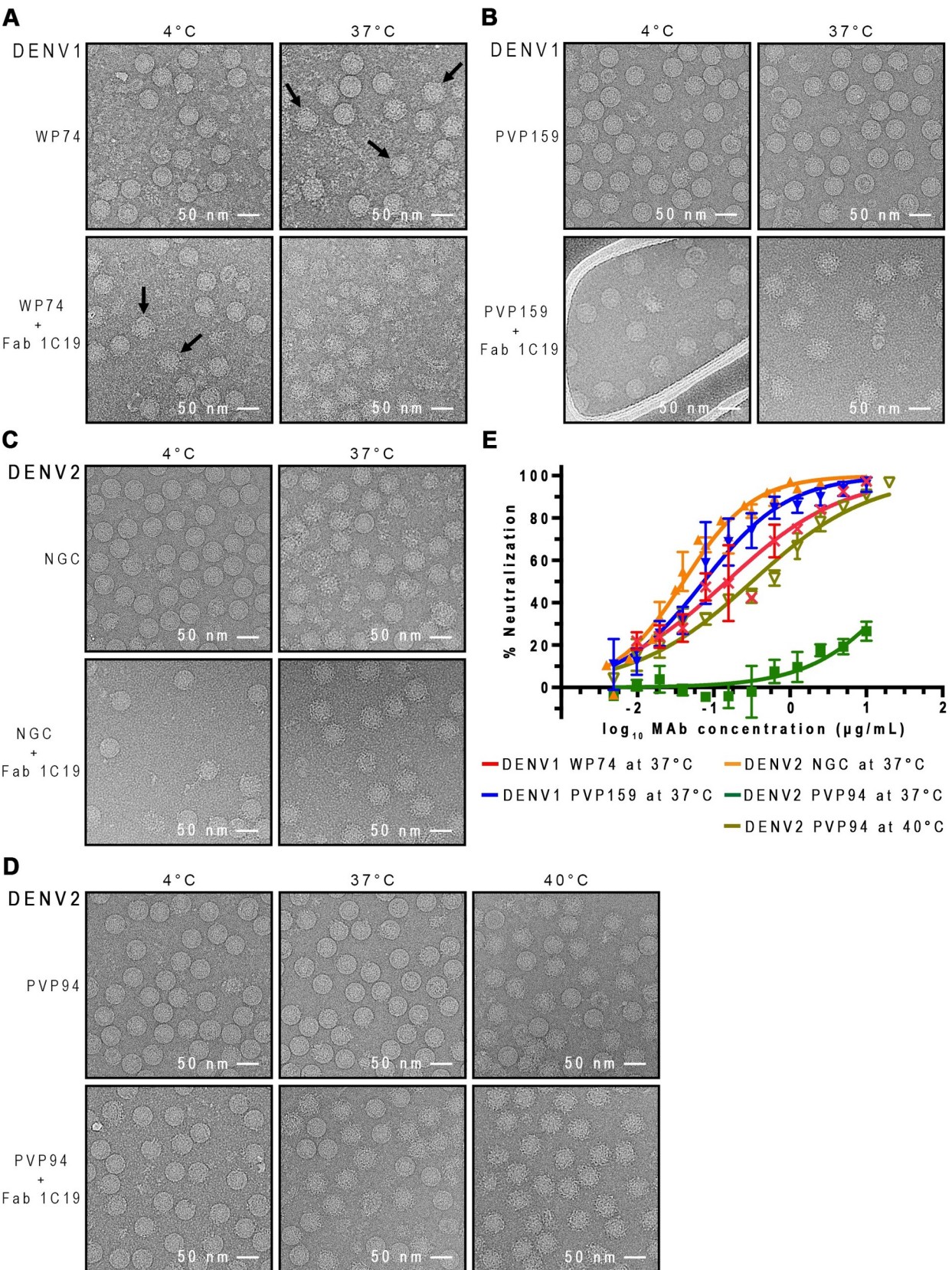

**Fig 1. HMAb 1C19 binding and neutralization for DENV1 is independent of the virus morphologies, whereas for DENV2 the activities depend on virus changing to bumpy-surfaced morphology, which likely leads to improved epitope accessibility.** CryoEM micrographs showing binding of Fab 1C19 to (A) DENV1 WestPac 74, (B) DENV1 PVP159, (C) DENV2 NGC, (D) DENV2 PVP94. All upper rows show the morphology of un-complexed viruses at different temperatures and the lower rows show the 1C19 complexes. For DENV1 WestPac 74, the un-complexed virus that became bumpy-surfaced at 37°C are indicated by black arrows. For all viruses, Fab 1C19 did not bind well to virus at 4°C, suggesting that the epitope was partially hidden. One example is shown when Fab 1C19 was added to DENV1 WestPac 74 at 4°C, only some particles formed complexes with the Fab as indicated by black arrows in (A) bottom left panel. At 37°C, Fab 1C19 bound to both DENV1 strains well even though the un-complexed DENV1 strains had different morphologies- WestPac 74 showing 50% particles with bumpy surface and PVP159 remains smooth-surfaced. The Fab-complexed particles are spiky and appear larger than the bumpy un-complexed virus. For DENV2 strains, the un-complexed NGC and PVP94 have a bumpy or smooth surface morphology, respectively. It appears that Fab 1C19 binds well to the bumpy-surfaced strain NGC, but poorly to the strain PVP94, as observed by 50% spiky particles in the PVP94 + Fab 1C19 sample. Increasing the temperature to 40°C increased the number of bumpy-surfaced particles in the un-complexed DENV2 PVP94 control leading to complete binding of Fab 1C19. (E) Neutralization profile of HMAb 1C19 for DENV1 or DENV2 strains. HMAb 1C19 neutralized DENV1 strains WestPac 74 and PVP159 equally well, whereas for DENV2, the antibody neutralized only strain NGC but not PVP94/07. HMAb 1C19 neutralized DENV2 strain PVP94/07 only at 40°C. The experiment was repeated three times. The neutralization profiles correlated well with the binding properties of Fab 1C19 observed in cryoEM.

strains, respectively) (Fig 2A–2D). To characterize the oligomeric state of the E protein used in BLI, a similar concentration of E protein (9 $\mu$g/mL) is used for glutaraldehyde cross-linking experiments. It is not possible to use Coomassie blue stained gel to visualize E protein bands as the concentration of E proteins is too low. Therefore, we conducted a western blot assay with HMAb 1C19 (which has been shown to be able to bind both monomeric and dimeric E protein on gel (S2A Fig, left)). The result showed that the oligomeric state of E proteins at 9 $\mu$g/mL primarily exists in a monomeric state (S2B Fig).

## HDXMS shows 1C19 epitope is mainly located at the *bc*, *hi* and *ij* loops on DII of the recombinant E protein

The rough boundary of the 1C19 epitope was identified to peptide resolution using HDXMS. This technique measures the Deuterium-for-Hydrogen (D-for-H) exchange rates of the amide groups on the protein backbone in $D_2O$ solution. The exchange rates depend on the protection factors of the amide groups. The protection factors are influenced by the structural features (secondary or tertiary structure) of proteins, which are stabilized through hydrogen bonding, disulfide bonds, electrostatic and hydrophobic interactions, and can change when the protein binds to ligands such as antibodies. Thus, HDXMS can provide information on structural differences at the binding site. The E protein alone or E protein-antibody complex was incubated in $D_2O$ sample buffer to allow D-for-H exchange for certain timings, followed by the addition of acid to quench the reaction. The samples were then digested to peptides by pepsin protease

**Table 1. HMAb 1C19 half maximal inhibitory concentration ($IC_{50}$) values in plaque reduction neutralization tests for DENV1 or DENV2 strains under conditions associated with differing particle morphologies.**

| Serotype | Strain | 1C19—rE protein binding affinity ($K_D$, nM) | Properties of viruses and antibody function at indicated temperature | | | |
|---|---|---|---|---|---|---|
| | | | 37°C | | 40°C | |
| | | | Un-complexed virus morphology | Neutralization $IC_{50}$ ($\mu$g/mL) | Un-complexed virus morphology | Neutralization $IC_{50}$ ($\mu$g/mL) |
| DENV1 | PVP159 | 0.712 | Compact, smooth-surfaced particles | 0.085 | | |
| | WestPac 74 | 0.844 | 50% of the particles becomes bumpy-surfaced | 0.15 | | |
| DENV2 | PVP94/07 | 19.015 | Compact, smooth-surfaced particles | Non-neutralizing | 60% of the particles becomes bumpy-surfaced | 0.32 |
| | NGC | 4.549 | Mostly bumpy-surfaced particles | 0.04 | | |

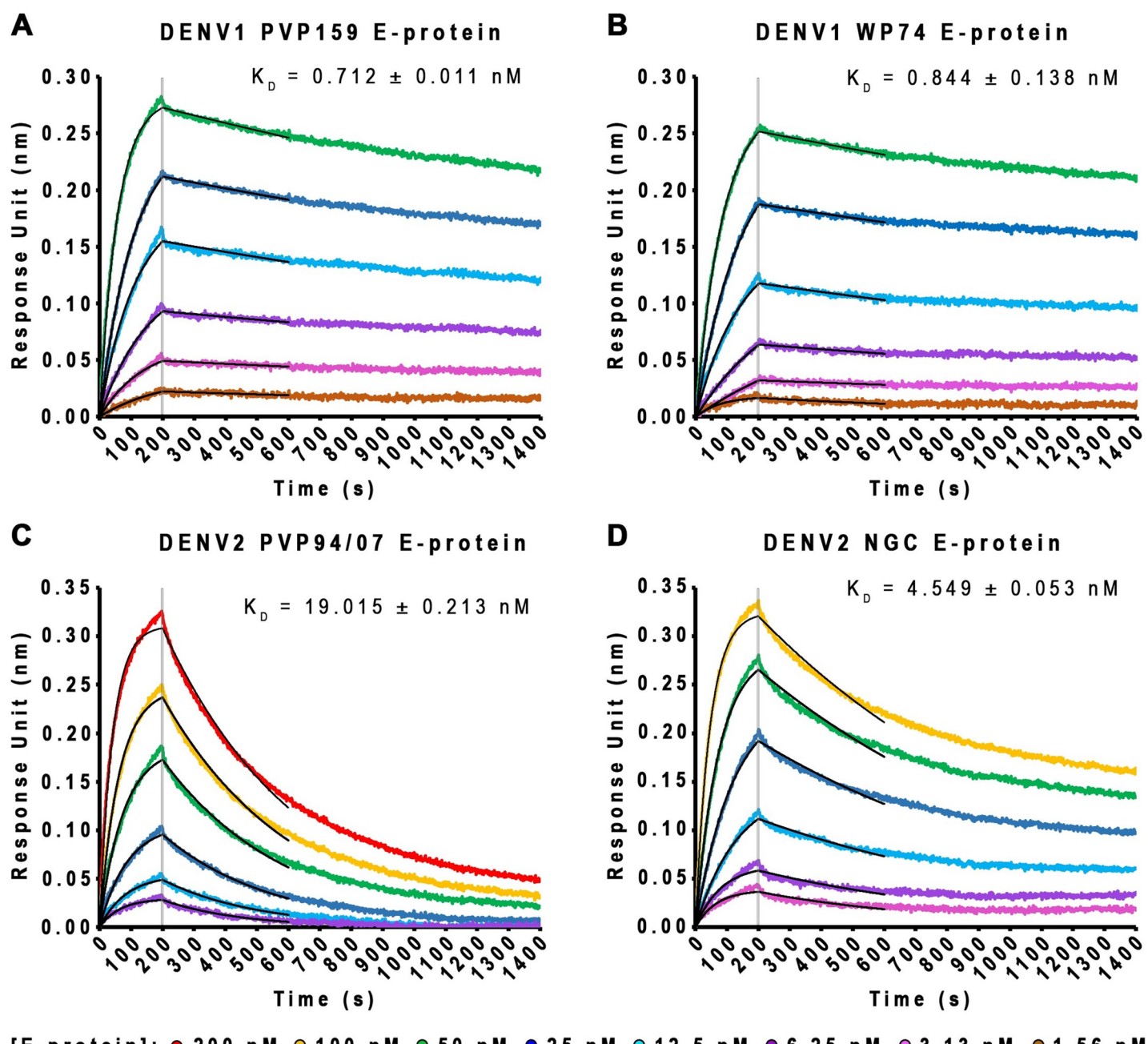

**Fig 2. HMAb 1C19 showed higher affinity of binding to E protein of DENV1 than to that of DENV2.** (A-D) Bio-layer interferometry curves (in different colors) and fitting curves (in black) for binding of increasing concentrations of DENV1 or DENV2 rE protein to immobilized IgG 1C19. These results showed HMAb 1C19 has a higher affinity of binding to DENV1 than to DENV2 E protein.

(see Methods section for details). The peptides were then analyzed by mass spectrometry to detect for the changes in the mass of the deuterated sample. The amounts of deuterium on peptides from a deuterated E protein alone sample were subtracted from those on similar peptides from deuterated E protein-antibody complexes (designated here as "deuterium alterations") for all time points tested (1, 2, 5, or 10 min). Several peptides were observed to have moderate to high deuterium alterations. Peptide 219–243 showed the highest deuterium

alteration of >1 Da at both 5 and 10 min incubation times (Fig 3A and 3B). This finding suggests that this region likely interacts directly with the Fab molecule. The second highest deuterium alteration group of peptides (~0.8 Da) includes peptide 91–108 and 236–251 (Fig 3A and 3B), suggesting that the Fab molecule may cover this region. Analysis of the surface charges of peptides comprising of the peptides with high deuteration alteration on DII showed charge complementary to the HMAb 1C19 complementarity-determining region (Fig 3C).

Peptide 302–315 had a lower level of deuterium alteration of ~0.65 (Fig 3A and 3B). This finding suggests the binding of the Fab 1C19 may be very dynamic, such that Fab binding to this region could be transient. This peptide is likely located on DIII of the opposite E protein protomer within a dimer. This is because this peptide is located too far away from the primary binding site (peptide 219–243 with the highest deuterium alteration) on DII within a protomer. To investigate if there is indeed presence of E protein dimers in the HDXMS mixture, we prepared E protein at a concentration of 3 mg/mL (similar to that used for HDXMS) to conduct glutaraldehyde cross-linking experiments. Both Coomassie-stained SDS PAGE and also western blot analysis using 1C19 antibodies showed the presence of monomers, dimers and higher order oligomers (S2A Fig).

Peptide 219–243 (*h* strand, *hi* loop and *i* strand), which is located at the inter-dimer interface on the virus surface, showed the highest level of deuterium exchange alteration among all peptides at the 10 min incubation timing (~1.25 Da) (Fig 3A and 3B). Other shorter peptides (223–235, and 224–235) within residues 219–243 and also the overlapping peptide (212–226 and 212–229) exhibited much less deuterium exchange alteration (0.5 to 0.6 at the 10 min incubation) suggesting that the entire length of residues 219–243 is optimal for antibody binding.

Peptide 236–251 (*i* strand, *ij* loop and *j* strand) is located at the E intra-dimer interface and it would be partially hidden on the compact smooth-surfaced virus structure (Fig 3B). The deuterium alteration profile of this peptide differed from that of peptide 219–243. At the 1 min time point, the deuterium exchange alterations were ~0.7 Da, which increased to ~0.85 Da at the 2 min incubation, but then gradually decreased to 0.65 Da and 0.55 Da at the 5 and 10 min incubation times, respectively. This finding suggests that Fab 1C19 has a lower affinity of binding for this part of the epitope, and therefore the larger magnitude of deuterium alteration is detected only at the shorter time points. The detachment of antibody from this peptide due to a low affinity of binding allows more deuterium to be incorporated into the E protein at longer incubation time.

Peptide 91–108 (c strand and *cd*-loop/fusion loop) is located adjacent to the *bc*-loop. Previous mutagenesis studies suggested that residues R73, G78 and E79 on the *bc* loop affect antibody binding [33] (Fig 3B). Unfortunately, HDXMS experiment did not obtain any peptides covering this *bc*-loop region and therefore we are unable to determine if this also forms part of the epitope. The previous study showed residues on the fusion loop (W101, G106 and L107) are not involved in the interaction with HMAb 1C19 [33], indicating that the fusion loop is not part of the epitope. Although the *cd*-loop may not form part of the 1C19 epitope, we observed some deuterium alternation in a peptide that encompassed this region (peptide 91–108) suggesting neighboring residues may be involved in antibody binding.

In conclusion, the HDXMS data suggested that HMAb 1C19 likely binds to the middle section of DII consisting of the *hi* loop.

## Cryo-EM structures of DENV complexed with Fab 1C19

Since the neutralization test results showed that HMAb1C19 neutralized DENV1 strains with equal potency, we chose to determine only the Fab 1C19:DENV1 WestPac 74 complex

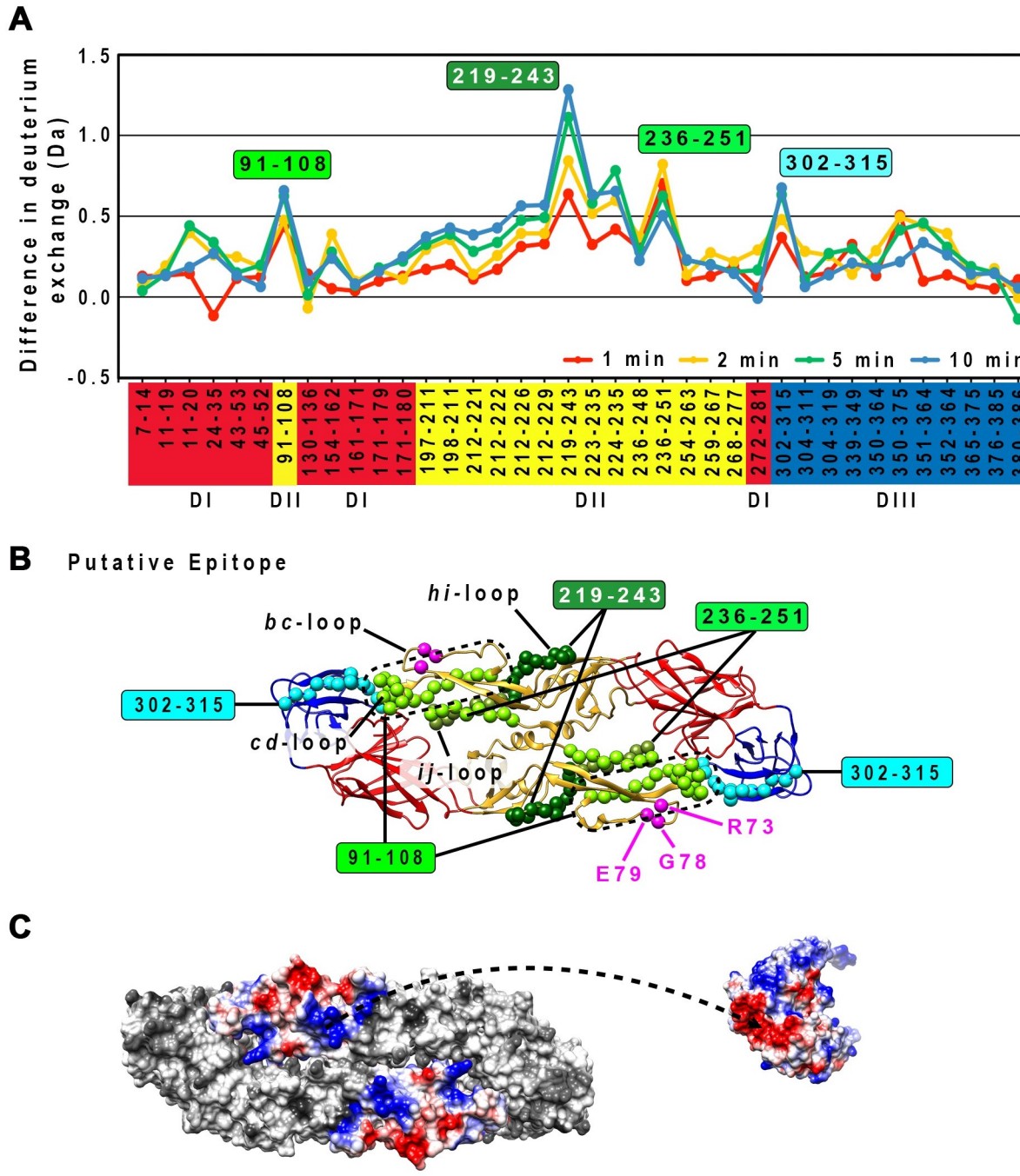

**Fig 3. HDXMS experiments showed that the 1C19 epitope contains residues in peptides 91–108, 219–243 and 236–251 on DII.** (A) Plot of the deuterium exchange differences between pepsin-generated peptide fragments of the un-complexed rE protein and the corresponding peptides from rE protein-Fab 1C19 complex at different incubation timings 1 (red line), 2 (yellow line), 5 (green line) or 10 (blue line) mins. Higher deuterium exchange difference indicates that the peptide has lower solvent accessibility when bound by Fab 1C19. Each dot on the plot represents a peptide fragment of the rE protein, as indicated on the x-axis. The domain where the peptide, on the x-axis, resides on the E protein is indicated by its box color: red, yellow or blue for DI, DII or DIII, respectively. Difference in deuterium exchange (deuterium alterations) by > 1 Da was considered significant, whereas between 0.5–1 Da was moderate. The peptides that showed moderate alterations may represent sites on the E protein that had weak binding or had undergone distant conformational changes upon antibody binding. Based on their deuterium alterations, peptides can be assigned into three groups. The first group is for peptides that showed significant alteration in deuterium exchange (peptide 219–243, dark green filled box). The second group consists of peptides with the second highest deuterium alteration, peptides 91–108 and 236–251, light green filled box. The third group contains other peptides with moderate deuterium alterations (peptide 302–315, cyan filled box). (B) Location of the peptides with high or moderate deuterium alterations on the E protein structure. The putative epitope consisting of residues in peptides 91–108 (light green), 219–243 (dark green) or 236–251 (light green) are shown as spheres

on the rE protein structure. Overlapping residues between peptides 219–243 and 236–251 are colored as olive-green spheres. DI, II or III of E protein are colored in red, yellow or blue, respectively. Residues that had been shown by site-directed mutagenesis studies to be important for HMAb 1C19 binding [33], are shown as magenta spheres. Peptide 302–315 is shown as cyan spheres. (C) The surface charges of peptides 91–108 (c strand and *cd* loop), 219–243 (*hi* loop), 236–251 (*ij* loop) in an E protein dimer are shown on the left, while the variable region of Fab 1C19 is on the right. Positive, neutral or negative charges are shown in blue, white or red colors, respectively. Charges of the putative epitope on the E protein and Fab seemed to be complementary to each other (arrow dashed line).

structure. At 37˚C, the HMAb 1C19 neutralized DENV2 strain NGC but not DENV2 strain PVP94. We did not try to solve the structure of the Fab 1C19 complexed with NGC, as we observed high heterogeneity of the particles. This observation was unsurprising, because even the un-complexed DENV2 strain NGC particles are structurally highly heterogenous [12], and Fab 1C19 binding may further increase particle heterogeneity. We decided to determine the structure of Fab 1C19 complexed with DENV2 PVP94 at 37˚C or 40˚C, where the antibody is non-neutralizing or neutralizing, respectively.

The DENV1 strain WestPac 74:Fab 1C19 complex showed two classes of particles; Class I or II groupings contained 26 or 49% of the virus populations, respectively (Fig 4C, right panel). Both cryo-EM maps were determined to ~19Å resolution (Figs 4A–4C, S3A and S3B), and the inability to achieve higher resolution suggested that the structure of the complex is flexible. The Class I DENV1-Fab 1C19 map showed clear Fab densities, and the Fabs bind to 60 of the 180 copies of E proteins on the virus surface (Fig 4A, left panel). Holes were observed near the base of the Fab densities that were located near the 3-fold vertices. A cross-section of the map showed that the E protein layer remained at a similar radius as the 28˚C unexpanded smooth-surfaced mature virus structure [14,34] (Fig 4A, right).

Although the Class II DENV1-Fab 1C19 map had a similar estimated resolution as the Class I map, the quality of the Class II map was poorer (Fig 4C). Compared to the Class I map, the E protein layer of the Class II particles had moved to a slightly higher radius (Fig 4A and 4C), similar to the structure of the DENV2 strain NGC at 37˚C [12] suggesting the E protein layer had loosened. There were holes located at the 3-fold vertices on the radii between 220–240 Å (the radii where E proteins are located in un-complexed DENV1), but additional densities were observed at higher radii of 240–280 Å in the same region, suggesting the E proteins had moved further outward. Protruding densities located adjacent to the 5-fold vertices showed the shape of Fabs, and therefore were interpreted as Fab 1C19.

To fit the E proteins and Fabs into their corresponding densities, we did translate and rotate some of the E proteins along the axis perpendicular to E protein dimer. The E protein and Fab models of the Class I cryo-EM map (Fig 4B) show that while the B-B′ dimer only moved very slightly from the original smooth compact virus structure, the mols A and C′ dimers are likely separated (Fig 5A and 5B). Hence, the solvent accessibility of the residues on mols A and C will be higher than in the former dimeric structure. The mols C are arranged underneath the B-B′ dimer, resulting in the presence of holes around the 3-fold vertices (Fig 5A, middle panel). The fitting showed that Fab 1C19 likely interacts with DII of mol A (Figs 4B and 5A). The epitope covered by the Fab molecule, as observed from the cryo-EM map of Class I particles (Fig 6A, left panel), overlaps with that identified by the HDXMS studies (Fig 6A, right panel). The low resolution of the Class I DENV1-Fab 1C19 map did not allow for precise analysis of the interacting residues. Electrostatic potential surface analysis showed complementarity between the cryoEM identified epitope and paratope (Fig 6B).

In the Class II cryoEM map of the DENV1 WestPac 74:1C19 complex (Fig 4C), the E protein shell seems to have moved outwards. The fitting of E protein mols A and C′ into the cryo-EM map of Class II particles was initiated by moving the E protein mols A-C′ dimer of uncomplexed DENV1 structure to a higher radius (240–280 Å). However, the density could not

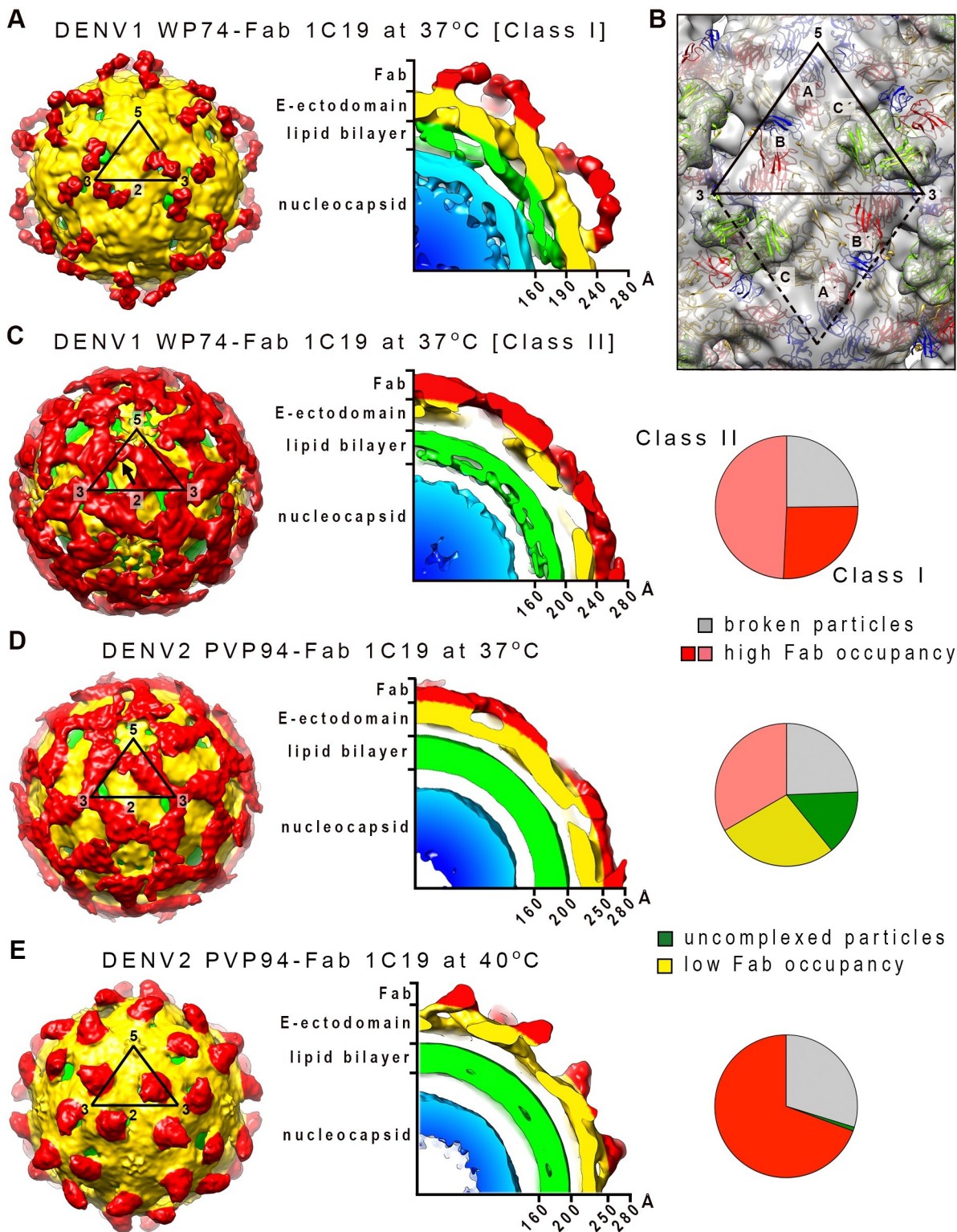

**Fig 4. The cryo-EM density maps of DENV1 strain WestPac 74:Fab 1C19 complex.** (A) Class I or (C) Class II particles at 37˚C, and DENV2 strain PVP94/07:Fab 1C19 complex at (D) 37˚C or (E) 40˚C. For all cryoEM maps, the surface (left) and center cross-section (center) are shown. (A) The cryo-EM map of Class I DENV1 strain WestPac 74:Fab 1C19 complex particles showed 60 copies of Fab (red) bound to the virus surface (yellow). The black triangle represents an icosahedral asymmetric unit, with the 2-, 3- and 5-fold vertices indicated. (B) The fit of E proteins and Fab 1C19 (light green) molecules into the Class I DENV1 strain WestPac 74:Fab 1C19 at 37˚C cryo-

EM map. All E protein and Fab molecule models were fitted into density, none were in negative densities and the molecules did not clash with each other. Three individual E protein molecules in an asymmetric unit are labelled as A, B and C, whereas the corresponding molecules in neighboring asymmetric unit within a raft are labelled as A′, B′ and C′. DI, II and III of E protein are colored in red, yellow and blue, respectively. (C) The Class II DENV1 strain WestPac 74:Fab 1C19 at 37˚C map likely has 60 copies of Fab molecules bound (black arrow). (D) Cryo-EM map of DENV2 strain PVP94/07:Fab 1C19 complex at 37˚C showed similarity with Class II DENV1 strain WestPac 74:Fab 1C19 complex at 37˚C, whereas the complex at (E) 40˚C appeared to be similar to Class I DENV1 strain WestPac 74:Fab 1C19 complex at 37˚C. In the rightmost panel of (C), (D) and (E), the proportion of the different particles within the virus population (right) is shown as a pie chart.

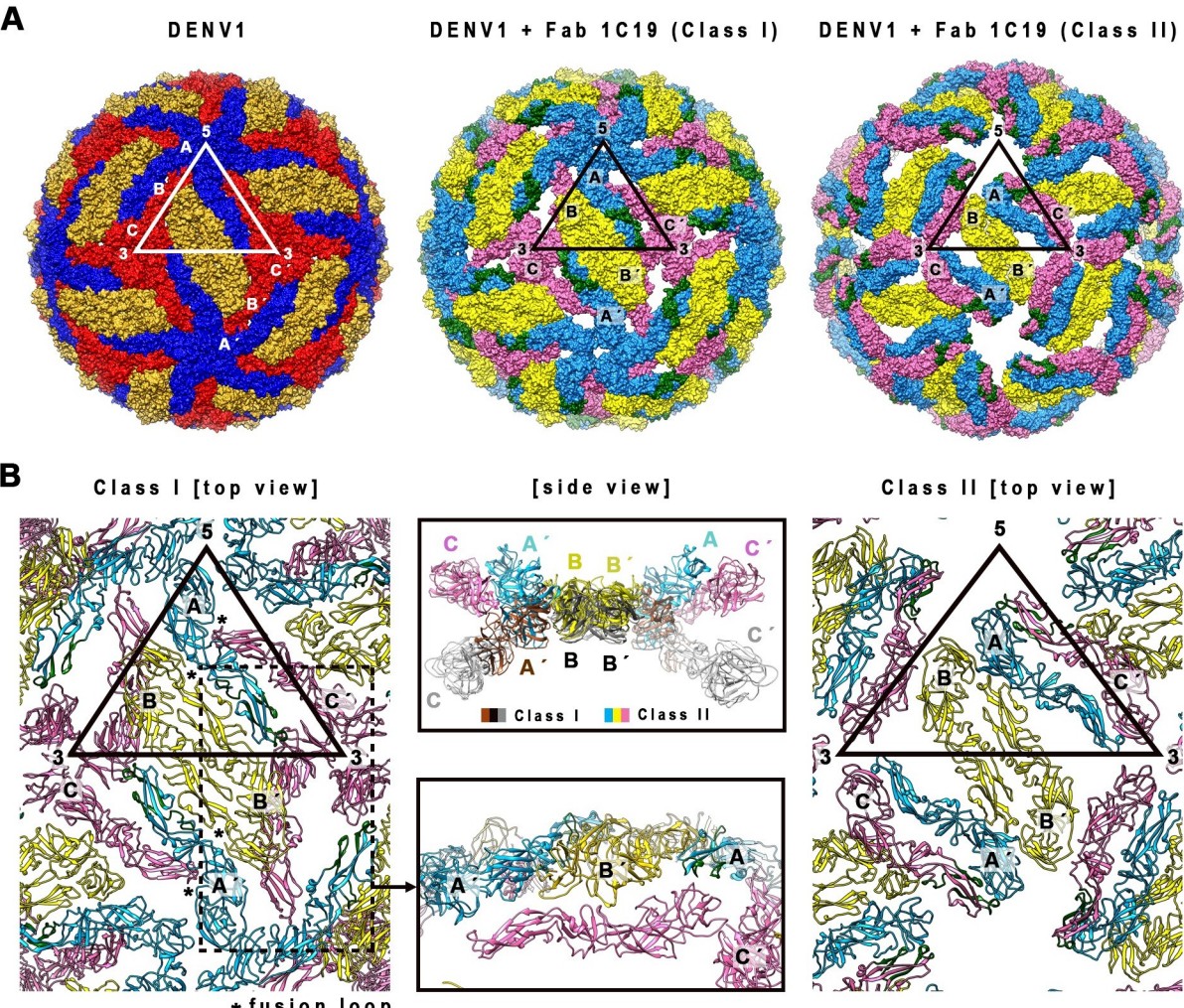

**Fig 5. E protein quaternary structure changes in the DENV1 strain WestPac 74-Fab 1C19 Class I and II structures compared to the un-complexed compact smooth DENV1 structure.** (A) E protein molecules organization on the un-complexed DENV1 (left) and the DENV1-Fab 1C19 complex Class I (middle) and II structures (right). The E protein molecules are shown as surface representation. In the un-complexed compacted smooth surfaced DENV1 structure, mols A, B or C E proteins are colored in blue, gold, or red, respectively, whereas the same molecules of the DENV1-Fab 1C19 complexes are in lighter shades of the same color (light blue, yellow or pink). The putative 1C19 epitope identified by HDXMS on mol A or C in the Class I and Class II structures, respectively are colored in dark green. (B) The E protein arrangements in Class I (left) and II (right) of DENV1-Fab 1C19 complex structures. (*Left*) E protein arrangement in Class I structure. Mol C (pink) in the complex structure did not clash with the adjacent molecules and is located on a lower radius. (*Center bottom*), Side view of the zoomed-in area of the dotted box on the left. (*Center top*) Superposition of an E-protein raft of Class I structure onto that of Class II structure. The E protein mols A and C of Class II complex particles are located at higher radii than those of Class I complex particles, while the E protein dimer mols B-B′ is at the same radius.

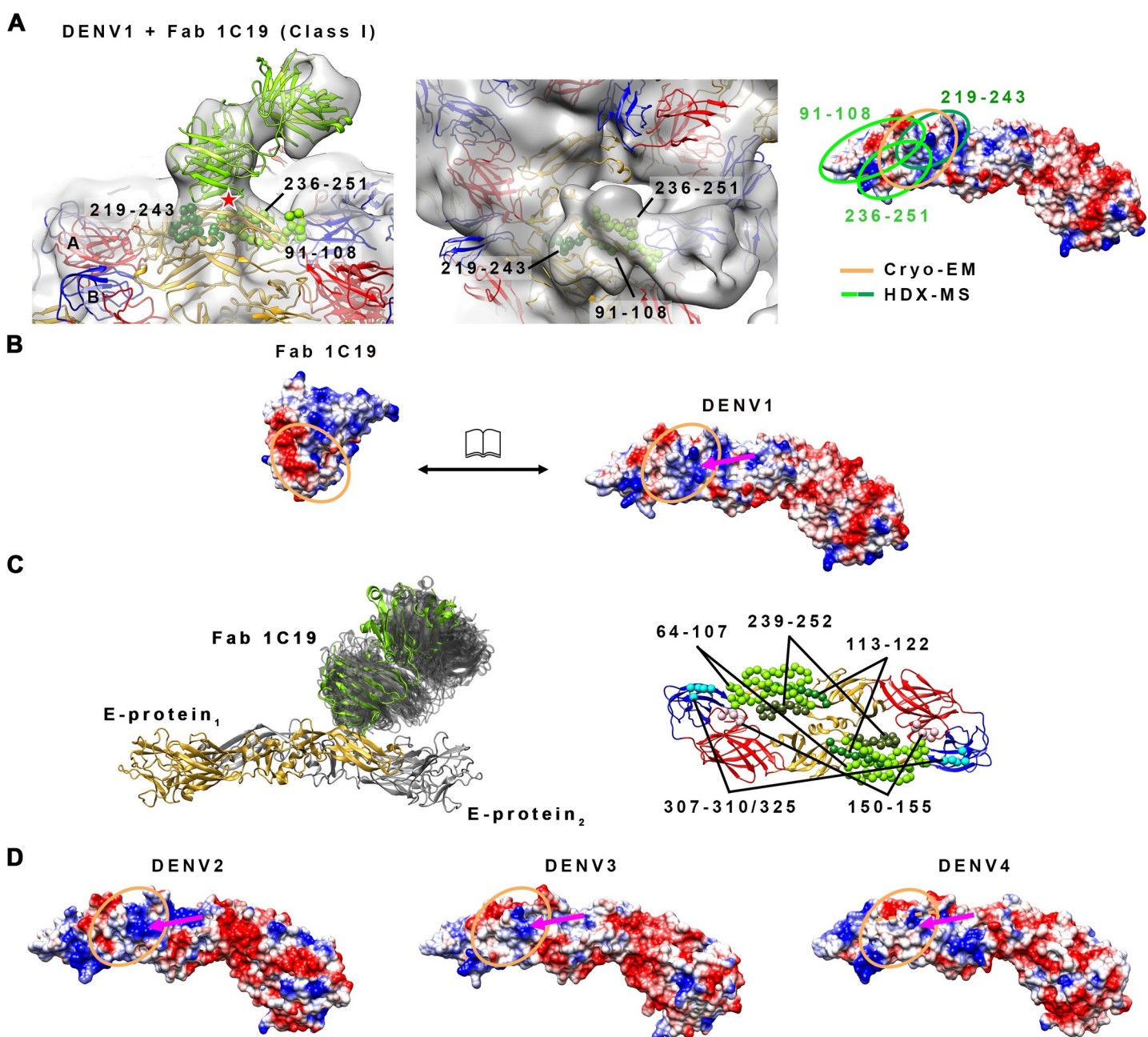

**Fig 6. The likely interactions between Fab 1C19 and DENV1 E protein, as suggested by combining HDXMS results, the cryo-EM structure and MD simulations, and its comparison to the other DENV serotypes.** (A) (*Left*) Zoom-in side view of the fit of Fab 1C19 molecules (light green) and E proteins in the cryo-EM map (gray transparent) of Class I particles. The putative epitope identified by HDXMS consists of residues in peptides 91–108, 219–243 and 236–251 (Fig 3B) are indicated by green spheres of different shades. The glycosylation at residue N67 is indicated by red star. (*Center*) Top view of Fab 1C19 density on fitted E protein. Fab 1C19 densities are located right on top of the putative epitope identified by HDXMS, mainly on peptides 219–243 and 236–251. (*Right*) The HDXMS results helped in guiding the fit of the molecule into the cryo-EM map, thus identifying the location of the 1C19 epitope on the E protein. The epitope size identified by HDXMS (light and dark green lines) is larger than that identified cryo-EM (orange circle). (B) The electrostatic charges on the interface surfaces of Fab 1C19 and DENV1 E protein (open book representation), identified by combining both techniques, are complementary to each other. Blue, white or red colors indicate positive, neutral or negative charges, respectively. The dashed orange lines indicate the border of heavy and light chains on the antibody paratope (left) and the corresponding footprint on the E protein (right). The magenta arrow indicates the positively charge patch on E protein that likely interacts with a negatively charged patch on the antibody paratope. (C) MD simulations reveal the highly dynamic properties of Fab 1C19 and its epitope on DENV1 E protein. (*Left*) Side view of E protein—Fab 1C19 complex. E protein dimer is shown in cartoon representation (grey and gold yellow colors) together with Fab 1C19 in its initial (green) near-experimental conformation. All visited states of C19 Fab over the entire simulation time are shown in transparent grey cartoon representation. (*Right*) The top view of E protein dimer shown in cartoon representation (domain I: red, domain II: yellow, domain III: blue) with the three distinctive regions (shown as spheres). (D) The electrostatic potential surface of the epitope on the other DENV serotypes. The epitope on DENV2 and DENV3 has a similar positively-charged surface (magenta arrow) as that on DENV1, while the same region on DENV4 is the least positively

charged surface. This finding is consistent with the neutralization profile of HMAb 1C19, which showed potent neutralization of viruses of the DENV1-3 serotype but not the DENV4 serotype. DENV structures used for electrostatic potential surface analyses were the cryo-EM structures of DENV1, DENV2, DENV3 or DENV4 (PDB codes 4CCT, 3J27, 3J6S or 4CBF, respectively).

accommodate the entire E protein dimer as a rigid body, especially for DI and DIII of E protein mol A (S4A Fig) suggesting that the interaction between the E protein protomers within the dimer had changed. We moved the E protomers separately, and the fit showed that their volume correlated well to the cryoEM map densities. The protruding densities conjoining to that of the E protein mols A-C′ dimer densities, which are adjacent to 5-fold vertices (Figs 4C and S4B), have an Fab-like shape. Since we had identified the relative position of the Fab to the E protein in the fitted Class I structure, we used this E-Fab structure to superimpose onto E proteins mols A and C′, in order to model these E proteins when complexed with Fabs. The Fab binding to E protein mol C′ showed nice complementary shapes and fit well into the density. For the Fab bound to mol A, part of it was consistently located outside of the densities (S4B Fig). This observation suggests that Class II particles are likely to have 60 copies of Fab 1C19 molecules bound (S4C Fig), similar to Class I particles, but the Fabs bind to E protein mol C.

The 2D averages of the DENV2 PVP94/07-Fab 1C19 complex particles at 37˚C showed heterogenous virus populations, with 15, 28, or 33% of virus particles with very low, partial or high Fab occupancies, respectively (Figs S1B and 4D, right panel). We reconstructed those particles with higher occupancies, indicated by the presence of spikes throughout the virus surface as observed in the 2D class averages (S1B Fig). The resultant cryoEM map (Fig 4D) is similar to the Class II map of DENV1 strain WestPac 74 at 37˚C (Fig 4C). Overall, the cryoEM map of the complex at 40˚C (Fig 4E) looks similar to that of the Class I DENV1 WestPac 74:Fab 1C19 map (Fig 4A). However, the binding orientation of the Fab molecules in the DENV2 strain PVP94/07-Fab 1C19 complex map differs, and the E protein layer is located at a higher radius (Fig 4E). Both cryoEM maps of the DENV2 PVP94/07-Fab 1C19 complex were determined to ~23 Å resolution (S3C and S3D Fig). The qualities of the density maps are lower compared to the DENV1 WestPac 74 complex maps. Due to this difference, the two maps were not fitted with E protein and Fab models.

## E protein–Fab 1C19 interactions observed using MD simulations

In order to explore the interaction between the DENV1 E protein epitope and the 1C19 Fab, we also employed explicitly solvated, atomic-resolution MD simulations. An initial conformation was first derived from a raw fit of the complex between an E protein dimer and 1C19 to the cryo-EM map, followed by refinement of the protein-protein interface by docking using the Rosetta software suite of programs. A pose with an RMSD of <4 Å to the starting position was identified within the most favorable 1% of observed energies across all complexes. This pose was used as the starting position for an MD simulation totaling 1.1 μs (see further details in Methods). During the course of the simulation, the Fab was observed to be dynamic with respect to the E protein surface (Fig 6C, left panel) indicated by the fluctuating angle of the Fab and the plane of the E protein dimer (S5A Fig). In Fig 6C, left panel, the visited states of Fab 1C19 on the E protein are shown across the entire trajectory. The dynamics of Fab 1C19 filtered by the extremes of its two most dominant motions throughout the trajectory (~80% of the total variance) are illustrated in S1 and S2 Movies. We also analyzed the per-residue E protein-Fab 1C19 contacts, which revealed that the E protein epitope could be divided into three regions around: i) residues 64–107, 150–155, and 113–122; ii) residues 239–252; and iii) residues 307–310 and 325 (Fig 6C, right panel). Thus, this analysis helps to resolve the transient

interactions between the E protein and Fab captured via HDXMS; furthermore, in the case of iii), the highly dynamic 1C19 Fab occasionally interacted with DIII of the neighboring E protein protomer within the dimer (Fig 6C, right panel, and S1 and S2 Movies) as also observed by HDXMS (Fig 3A). However, the MD simulations of the soluble E protein dimer did not show dissociation, as observed in the context of the whole virus by cryoEM. The same simulation protocol applied to the DENV2 NGC—Fab 1C19 complex showed an even more dynamic Fab (S5B Fig), interacting with approximately only half the number of residues when compared to DENV1 (S5C Fig).

## Discussion

HMAb 1C19 binds to virus strains of all four DENV serotypes but only neutralizes DENV1-3 [33]. Comparison of the surface charges of the epitope on all DENV serotypes showed that the charge profiles are largely similar (Figs 6B and 6D and S6), consistent with the ability of 1C19 to bind to all DENV serotypes. The DENV1 epitope is mostly positively charged, with some hydrophobic and negative patches. This positively-charged surface is complementary with a large negatively-charged surface on the Fab 1C19. The positively-charged surface can be found on the corresponding regions of DENV2-4 (Fig 6D). However, we also noted some surface charge differences between serotypes, for example, DENV3 has a negatively charged surface in the *hi*-loop region, whereas for DENV4, the epitope is less positively charged (Fig 6D). Comparing the electrostatic charges of the 1C19 epitope between serotypes showed that DENV4 differs the most in terms of charge, which may explain the inability of HMAb 1C19 to neutralize DENV4. We also generated a sequence alignment of the epitopes across DENV1-4 (S6 Fig) to show that DENV4 has unique residues in the following positions: residues 64 (corresponding residues in DENV1/DENV4 is K/S), 91 (V/I), 95 (T/D), 96 (F/V), 120 (K/L), 122 (V/S), 222 (S/A), 228 (Q/E), 229 (E/V), 236 (L/R), 242 (T/V), and 251 (V/T).

The epitope identified by HDXMS, cryoEM and MD correlates very well. MD simulations were performed over a timescale 1.1 μs (S5A Fig); the closest HDXMS condition to this, is the 1 min $D_2O$ incubation time (Fig 3A). MD identified residues 239–252 (Fig 6C), and the HDXMS at 1min $D_2O$ incubation shows that the highest altered deuterated peptide is between residues 236 to 251 (Fig 3A). The cryoEM fit also shows that E protein residues 236–251 are within the region where the Fab is bound (Fig 6A, middle).

The results generated by combining HDXMS, cryoEM and MD simulations suggest antibody binding to the epitope occurs in a very dynamic manner involving motions of both antibodies and the E protein. This finding could explain why only low-resolution cryoEM structures can be obtained, even when the E protein layer did not move to a higher radius, as observed in the Class I DENV1 WestP74-Fab 1C19 at 37˚C map (Fig 4A). This interpretation is also consistent with the observation that the antibody is unable to bind to virus at 4˚C and, in general, we observed an increased efficiency of antibody binding with increased incubation temperature.

The MD and HDXMS results indicate that there are significant dynamic properties of the virion particle affecting Fab 1C19 binding to E protein, with the majority of the epitope located on DII of a protomer, but also that the antibody can bind transiently to DIII on the opposite protomer. MD simulations show that the constant region of Fab is involved in transient binding to DIII. This interaction likely does not form the main binding epitope. This is because binding of Fab across the E proteins requires the E protein to be in a stable dimeric structure; in this case, you would expect that the Fab would prefer to binding to virus at 4˚C where the dimers are the most stable. However, the opposite is observed, few or no Fabs are observed to bind to DENV1 or DENV2 particles at 4˚C. Also, in the cryoEM structure, the E protein

protomers within the dimer had dissociated after Fab binding, suggesting that the DII residues on one protomer alone are likely sufficient for binding. The epitope of HMAb 1C19, as determined by MD and HDXMS studies, would be partially hidden around the *hi-* and *ij-* loops on the virus surface. As mentioned above, HMAb 1C19 only binds to DENV at 37°C or higher. The E proteins intradimer and interdimer interactions on uncomplexed DENV2 NGC have previously been shown to be loosened at 37°C [12]. On the warm DENV, the 1C19 epitope is more exposed, allowing the binding of HMAb 1C19. HMAb 1C19 may further induce E protein dissociation due to the dynamic properties of this antibody (Fig 6C). It could be that Fab 1C19 "wiggles" the bound E protein out from the surrounding interactions. If the soluble truncated E proteins are present as dimers in solution (such as those proteins used in the HDXMS experiment and the MD simulations), the E protein may tolerate some level of "bending" between the E protomers without dissociating from each other—thus the E protein dimer dissociation caused by antibody is not observed. The HDXMS results also did not identify peptides with significant deuterium exchange on soluble E proteins along the intradimer interface on either side.

Since incubation of DENV at 37°C was important for binding of HMAb 1C19 to its epitope, we compared the neutralization profile of HMAb 1C19 to various strains of DENV1 or 2 that differ in their particle morphologies (*i.e.*, do or do not change to bumpy particles at 37°C) (Fig 1E and Table 1). For DENV1, HMAb 1C19 neutralized all DENV1 strains regardless of whether or not the surface of the virus can change in structure at 37°C (Fig 1A, 1B and 1E and Table 1). On the other hand, the ability of HMAb 1C19 to neutralize DENV2 depended highly on the ability of the virus to undergo structural changes at 37°C (Fig 1C–1E and Table 1). HMAb 1C19 exhibited a high level of neutralizing activity against the DENV2 strain NGC (whose un-complexed particles are bumpy at 37°C), but we did not detect neutralizing activity to DENV2 strain PVP94/07 (whose un-complexed particles are smooth at 37°C) (Table 1). The cryoEM micrograph of the DENV2 strain PVP94/07 particles incubated with Fabs showed ~60% DENV2 strain PVP94/07 particles turning spiky, suggesting there is some level of Fab binding. The 2D averages of these particles showed a range of particles with varying Fab occupancies (S1B Fig). These findings suggest poor binding efficiency, consistent with the inability of the HMAb 1C19 to neutralize this strain of virus at 37°C. When the temperature was raised to 40°C, the un-complexed DENV2 strain PVP94/07 becomes bumpy, suggesting a looser arrangement of E proteins. When Fab was added (Fig 1D), 2D class averages of the particles showed a shift towards a larger percentage of particles (Fig 4E) with higher Fab occupancies (S1C Fig) than those at 37°C (S1B Fig), hence allowing Fab 1C19 to neutralize (Fig 1E).

We considered whether this difference in neutralization could also be affected by differences in surface residues within the epitope between strains, leading to reduced antibody binding. To resolve this question, we conducted biolayer interferometry measurements to compare binding of HMAb 1C19 to recombinant E proteins of different strains within a serotype (Fig 2B). HMAb 1C19 binds to E proteins of DENV1 PVP159 and WestPac 74 strains at subnanomolar affinity (0.71 and 0.84 nM, respectively) (Fig 2A and 2B). Whereas the affinity to E protein of DENV2 PVP94/07 and NGC strain are lower at nanomolar range (19.02 and 4.55 nM, respectively) (Fig 2C and 2D). The affinity differences between DENV1 and DENV2 strains can be seen from the dissociation rate, which is steeper for E proteins of DENV2 strains. To understand why the affinity of 1C19 is lower for binding DENV2 E protein, we conducted MD simulations of DENV1 and DENV2 E proteins complexed with Fab 1C19. For DENV2 MD simulations, we used the cryoEM DENV1:Fab 1C19 complex structure but we substituted the E protein sequence with that of the DENV2 NGC strain (S5C Fig). The simulations revealed that the Fab on DENV2 was more dynamic than the Fab on DENV1 (S5B Fig), and there are likely only 43 interacting residues on DENV2 DII E protein, about half of those on DENV1

(79 residues). Also, Fab 1C19 is not likely to have any interaction with DIII on DENV2 E protein (S5C Fig) compared to the transient interactions on DENV1 (Fig 6C). Since the affinity of HMAb 1C19 is generally poorer to DENV2 strains, the ability of HMAb 1C19 to neutralize for the DENV2 strain NGC better than DENV2 PVP94/07 at 37˚C is partly due to the bumpy surface morphology of NGC virus surface (Fig 1C) that increases the solvent accessibility of the epitope compared to the smooth surface morphology of DENV2 PVP94/07 (Fig 1D). In the case of DENV1, since the affinities of HMAb 1C19 to all DENV1 strains E proteins are high, the antibody can neutralize all strains regardless of its morphology at 37˚C (Fig 1A, 1B and 1E). However, we expect that there is some subtle motion of E proteins at 37˚C on the smooth unexpanded DENV1 PVP159 particles (Fig 1B). It is possible that the antibody is able to outcompete the E-to-E protein interactions to access its epitope when the affinity of the antibody is high, even if the epitope is partially hidden on the virus surface (for example in the compact smooth surface virus morphology).

The cryo-EM map of Class I DENV1 strain WestPac 74-Fab 1C19 complexes at 37˚C shared similarities with the map of the DENV2 strain PVP94/07-Fab 1C19 complex at 40˚C; however, the latter map showed the movement of the E protein layer to a higher radius. These data suggest movement of E proteins, therefore leading to the poorer densities of both the Fab and E proteins of the DENV2 strain PVP94/07-Fab 1C19 complex in the 40˚C cryoEM map. Judging from the location of the Fab densities, DENV2 strain PVP94/07-Fab 1C19 at 40˚C is similar to the Class I DENV1 WestPac 74-Fab 1C19 complex, although with a different orientation of the Fab. Most likely, the E protein arrangements in the two maps differ slightly.

The DENV1 WestPac 74-Fab 1C19 Class II and the DENV2 PVP94-Fab 1C19 structures at 37˚C are similar to each other, and they showed a different binding mode with Fab 1C19 compared to that in the DENV1 WestPac 74-Fab 1C19 Class I map (Fig 4C and 4D). The Fab 1C19 binds to the E protein mol C in DENV1 WestPac 74-Fab 1C19 Class II and the DENV2 PVP94-Fab 1C19 at 37˚C structures, whereas in the DENV1 WestPac 74-Fab 1C19 Class I, mol A is bound (Fig 5). These data suggest that the Fab 1C19 can only bind to either one of the protomers in the mol A/C′ dimer, depending on which is exposed first during the E protein movement at 37˚C, hence the two classes do not represent the sequence of occurrence but rather two individual states. On the other hand, the B/B′ dimer in all of our maps did not move much from the original radius in the compact smooth particles and Fabs did not bind to this dimer.

These structural findings provide a better understanding of how the antibody neutralizes the virus once it achieves sufficient occupancy on the virus surface. Part of the epitope of HMAb is located on *hi-ij* loops, which is near to the glycosylation site on residue N67 (Fig 6A). The N67 residue is important for binding to the DC-SIGN receptor [35], and hence HMAb 1C19 binding to the epitope may prevent this interaction. In addition, HMAb 1C19 may also block fusion. Comparison of the 4Å resolution structure of E protein of the mature DENV1 [36] to the crystal structure of the post-fusion DENV1 E protein showed that the *ij* loop, which is also part of the 1C19 epitope, has a very different conformation (S7 Fig). When the antibody binds to this loop, it may stop the structural changes needed to transition to the post-fusion structure, thus inhibiting the fusion process. In summary, we proposed models of how HMAb 1C19 could bind and neutralize viruses. (Fig 7). HMAb 1C19 mainly binds to a partially hidden epitope on E protein DII of either mol A or C–depending upon which is exposed first when the E proteins are "vibrating" at higher temperatures (37˚C or 40˚C). Once bound to the epitope, HMAb 1C19 can disrupt the quaternary structure of the virus, this may prevent the virus from attaching to the cell and/or blocking virus-endosomal membrane fusion.

*In vitro* work from Smith *et al*. (2013) [33] showed the neutralization capability of HMAb 1C19 to lab adapted virus strains in mammalian cells to be very potent against DENV 1–3.

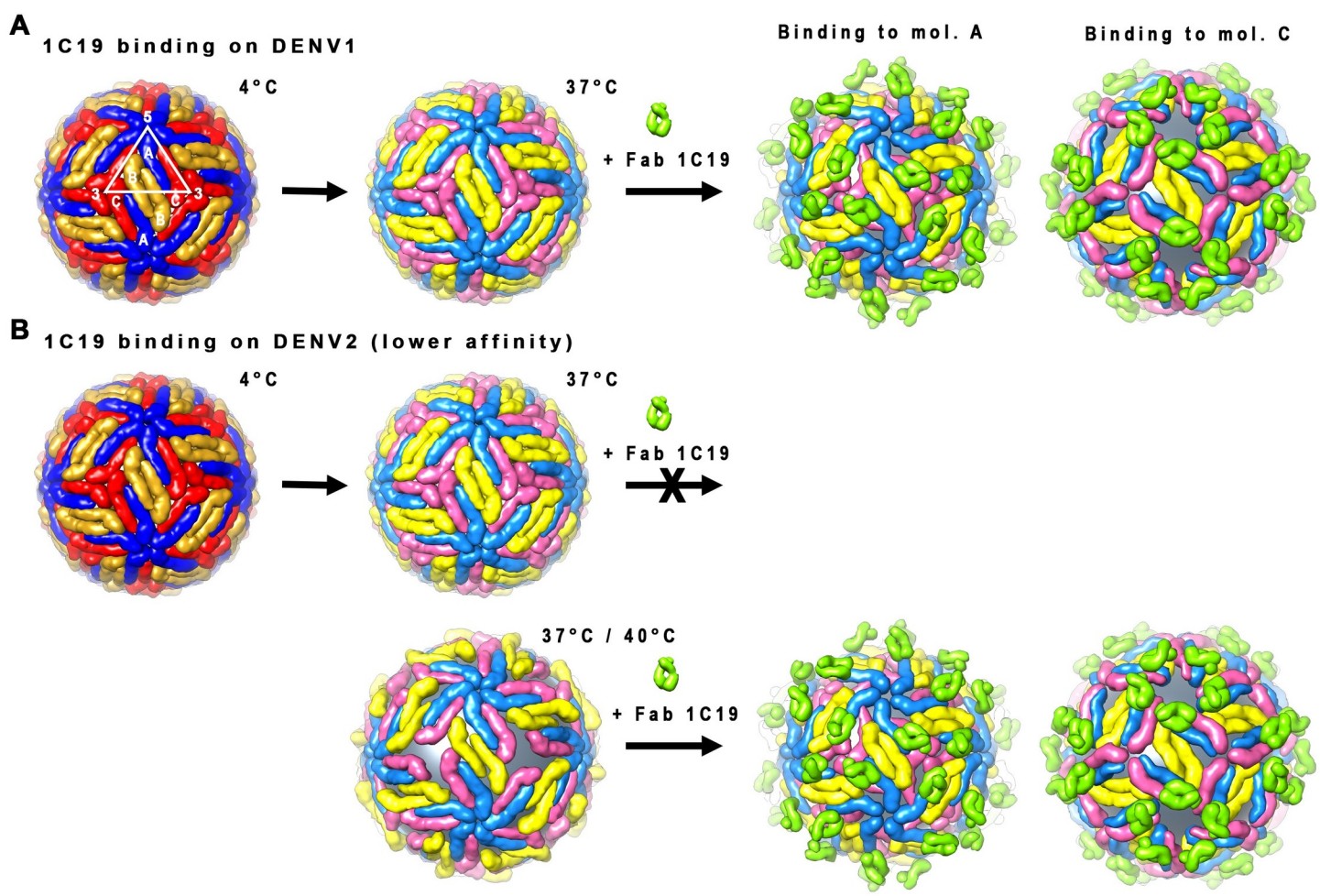

**Fig 7. Binding of Fab 1C19 to DENV1 and DENV2. Fab 1C19 binds to a partially hidden epitope on DII of the E proteins.** It has a strong affinity to DENV1 E protein and therefore it can out-compete E-to-E interactions even when the virus maintains a smooth compact surface that only partially reveals the epitope. Fab 1C19 can only bind to one of the E proteins of the A-C dimer, depending on which is exposed first when the E protein is vibrating at 37˚C. Fab 1C19 binds in a similar same way to DENV2 (either mol A or C), but because its affinity to DENV2 epitope is lower, the virus has to change its conformation to a looser "bumpy" surface structure at higher temperatures (37˚C or 40˚C) to expose its epitope fully before the antibody can bind.

When tested in a mouse model, it has a modest effect (1 log reduction) on DENV1 viremia. In another completely different system, where HMAb 1C19 is added to DENV from patient sera and then tested for its inhibitory activity to infect mosquitoes [37], this showed HMAb 1C19 is not neutralizing. These observations suggest that HMAb 1C19 is not a good therapeutic agent, consistent with our findings. However, the results from this study have important implications for the design of therapeutic monoclonal antibodies for prevention or treatment of DENV infection. Previous structural studies showed potent antibodies binding to the smooth surface of compact virus, which bind across E proteins to cause locking of the virus quaternary structure. In the work presented here, it is apparent that if an antibody has high affinity of binding to E protein monomers, even when the epitope is partially hidden, that antibody can neutralize virus in all morphologies. However, if the affinity of binding is lower, then the strength of binding depends on the transition to a virus morphology that exposes the epitope. Prophylactic antibodies would be used on individuals before infection, a scenario in which the antibodies would encounter virus at an average human body temperature of 37˚C. At this temperature,

there could be a mixture of virus morphologies–smooth- and bumpy-surfaced particles, depending on the virus strain. Thus, a target product profile might consider the role of virus morphologies in effectiveness. For example, one could use a high-affinity antibody that could bind to its epitope regardless of virus morphology as a preferred antibody molecule. Alternatively, it might be possible to use a cocktail of antibodies, each effective against a specific virus morphology. Many DENV-infected patients experience fever, therefore therapy might occur in subjects with elevated body temperature. In our current and previous studies [34] (Fig 1), incubation of un-complexed DENV at 40˚C seemed to increase the number of bumpy-surfaced virus particles. Therefore, antibodies that bind across E proteins recognizing a quaternary structure that is only preserved in the smooth compact surface particles might be rendered less effective at high temperature, and antibodies recognizing E protein monomers may be a more effective choice in that setting. The data shown here suggests that increasing the affinity of antibody recognition of E protein monomers may make an antibody more robust for binding to differing virus morphologies.

## Methods

### Recombinant E protein preparation

DENV1 PVP159 strain E protein ectodomain (residues 1–394) coding sequence was cloned in vector pMT/BiP/V5-HisA (Thermo Fisher) followed by transfecting the plasmid into *Drosophila melanogaster* Schneider 2 cells (Life Technologies). DENV1 WestPac74 strain and DENV2 PVP94/07 and NGC strains E protein ectodomain (residues 1–394) were cloned downstream of an IL2 leader sequence in the pcDNA 3.4-TOPO (Invitrogen) and then the plasmid was transfected into EXPI293 cells (Thermo Fisher). The recombinant E protein was secreted in supernatant and the supernatant was clarified by centrifugation at 9000 x g at 4˚C for 30 min. The rE protein was purified using an immobilized dengue-specific antibody 4G2 affinity column. The bound rE protein was eluted with 0.1 M glycine-HCl, pH 2.7. The rE protein containing fractions were immediately neutralized by adding 1 M Tris-HCl at pH 9.0 and buffer-exchanged to 20 mM Tris-HCl pH 7.5, 150 mM NaCl.

### Epitope mapping by hydrogen-deuterium exchange mass spectrometry (HDXMS)

To achieve complete binding of the epitope, 10 $\mu$L of DENV1 rE protein (4.8 mg/mL) was mixed with 10.1 $\mu$L of Fab 1C19 (8.6 mg/mL) to achieve a molar ratio of 1.5 Fab molecule to every E protein (50% excess) and incubated at room temperature for 1 h. E protein (without Fab 1C19) in sample buffer (10 mM Tris-HCl, 150 mM NaCl, pH 7.5) was used as a zero deuterium exchange control. To prepare $D_2O$ sample buffer, first the water of the sample buffer was evaporated using a centrifugal vacuum concentrator, and then 99.9% $D_2O$ water was used to resuspend the remaining sample buffer salt. The hydrogen-deuterium exchange reactions were initiated by diluting 2 $\mu$L of E protein-Fab1C19 complex into 18 $\mu$L of $D_2O$ sample buffer, resulting in a final concentration of ~90% $D_2O$. The protein samples were incubated at room temperature for 1, 2, 5 or 10 min. The exchange-reaction was stopped by the addition of 30 $\mu$L of pre-chilled quench solution containing 0.067% (v/v) TFA, 2.5 M guanidium hydrochloride and 8.3 mM DTT, which lowered the pH to 2.5. The quenched sample with a total volume of 50 $\mu$L was subjected to an online pepsin digestion using Waters enzymate BEH pepsin column (Waters, Milford, MA) and then was injected into a chilled nanoACQUITY UltraPerformance LCs (UPLC) system (Waters Corporation, Milford, MA). Mass-spectrometry data acquisition was done as previously described [38].

ProteinLynx Global SERVER software (PLGS v2.4) (Waters Corporation, Milford, MA) was used to identify the pepsin-generated peptides from non-deuterated control samples. This set of peptides then was verified further using the Dynamx 2.0 (Waters Corporation, Milford, MA, USA). The full list of peptides then was manually examined by searching the MS scan of a non-deuterated protein sample to test for correct m/z state and check for the presence of overlapping peptides. The differences in the average masses of the non-deuterated and deuterated E protein alone peptides showed the surface exposed regions of the E protein. While the differences between deuterated E protein and that of the E-Fab 1C19 complex peptides showed the footprint of antibody and also the induced structural changes of the E protein upon antibody binding. The deuterium exchange differences between 38 peptides of the deuterated E protein and the E protein–Fab 1C19 complex then were plotted. This testing represents ~61% of the total primary sequence of DENV1 E protein.

## Virus sample preparation for cryo-EM studies

Four DENV strains were used in this study: DENV1 strain WestPac 74, DENV1 strain PVP159, DENV2 strain NGC and DENV2 strain PVP94/07. The virus was propagated and purified as described previously [13]. The purified virus was concentrated, and buffer exchanged into NTE buffer (12 mM Tris-HCl pH 8.0, 120 mM NaCl and 1 mM EDTA) using Amicon Ultra-4 centrifugal concentrator (Millipore). All purification procedures were done at 4˚C. The purity and protein concentration of the purified virus were estimated by SDS-PAGE after Coomassie blue staining. The E protein concentration was estimated by comparing the corresponding band with a series of bovine serum albumin protein standards of known concentration.

## Cryo-EM sample preparation and data acquisition

Purified viruses were incubated with Fab at three different temperatures (4, 37 or 40˚C) for 30 min with a molar ratio of one Fab for every E-protein and followed by incubation at 4˚C for 2 h. The DENV controls (without Fab) also were prepared in the same way. A 2.5 μL sample was applied onto ultra-thin carbon-coated lacey carbon grid (Ted Pella) and blotted with filter paper for 2 s before quick plunging into liquid ethane using the FEI Vitrobot Mark IV. Frozen grids were kept at liquid nitrogen temperature. Image acquisition was done on a 300 kV— Titan Krios cryo-electron microscope (FEI company) operated at nominal magnification of 47,000X and an electron dose of 18 e⁻/Å². The images were recorded on a direct electron detector (Falcon, FEI) with effective pixel sizes of 1.71 Å/pixel. Images showing strong drift and astigmatism were discarded. For DENV1 strain WestPac 74-Fab 1C19 complex at 37˚C, a total of 475 images with a defocus range of -1.0 to -3.7 μm were selected for further processing. Whereas for DENV2 strain PVP94/07-Fab 1C19 complex at 37˚C or 40˚C, 722 or 772 images, respectively, with defocus range of -0.5 to -4.3 or -0.8 to -4.6, respectively, were selected.

## Cryo-EM image processing and reconstruction of DENV1 WestPac 74 –Fab 1C19 complex at 37˚C

A total of 4,301 spiky-looking unbroken DENV1-Fab 1C19 complex particles was selected manually using *e2boxer* tool in the EMAN2 software package [39]. The contrast transfer function (CTF) parameters were estimated by using *fitctf*, and then manually verified using the *ctfit* program in EMAN [40]. Orientation search of all particles was done using Multi-Path Simulated Annealing (MPSA) [41] and three-dimensional reconstruction was done using the *make3d* from EMAN. Mature DENV1 cryo-EM map [36] was used as an initial model. The final map was reconstructed from 1,116 particles. Resolution of the map was estimated to be ~19 Å, which was calculated by plotting the Fourier shell correlation coefficient between two

reconstructed maps of two half-datasets of the final iteration step with a cutoff value of 0.5 (S3A Fig). The correctness of the map was validated by two approaches. In the first approach, the dataset was split into two-halves, *i.e.*, odd and even numbered particles. The two-halves datasets were reconstructed separately using mature DENV1 structure as initial model. In a second approach, a model generated from starticos [EMAN [40]] was used. The reconstructed maps obtained were similar to that of the final DENV1-Fab 1C19 map.

The particles selected in the final iteration step of the cryo-EM map in the reconstruction mentioned above represent only ~26% of the total population. Further investigation of the structures of the complex particles in the remaining particles population was carried out by Relion [42]. The particle coordinates were imported from EMAN and CTF parameters were estimated from the micrographs using *ctffind3* [43]. Two-dimensional classification was done to remove broken/irregular shape particles and further 3D classification was carried out with mature DENV1 28˚C cryo-EM map as initial model. Two classes were observed, Class I was similar to the reconstructed map mentioned above which has 60 Fab molecules bound to virus surface and Class II map which is slightly larger in diameter with a gap between the lipid bilayer and E protein layer. Further 3D refinement was done only for the Class II structure and it was carried out using Relion with Class II structure obtained from 3D classification as the initial model. The estimated resolution at the final iteration was ~25 Å. The resolution was estimated by plotting the Fourier shell correlation coefficient between two-half datasets independently reconstructed maps with a cutoff value of 0.5. The maps generated from the two-half datasets also showed high similarities suggesting the correctness of the reconstructed maps. The reconstruction steps using Relion were done with a dataset that was binned to a pixel size of 6.8 Å/pixel. To further improve the resolution, the orientation search and three-dimensional reconstruction were continued using MPSA and *make3d* with particle images in original pixel size of 1.7 Å/pixel. A total of 2,121 particles was selected in the final iteration step and the reconstructed map had a resolution of 19 Å (S3B Fig). The resolution was calculated using the same method used to estimate the resolution of the Class I map mentioned earlier.

## Cryo-EM image processing and reconstruction of DENV2 PVP94/07 –Fab 1C19 complex at 37˚C and 40˚C

The CTF parameters for each micrograph were estimated using ctffind3 [43]. A total of 38,865 and 37,568 particles were selected from DENV2 strain PVP94/07-Fab 1C19 complex at 37˚C and 40˚C samples, respectively, using e2boxer. Two-dimensional classification was done to remove broken/irregular shaped particles. The 2D class averages of DENV2 PVP94/07-Fab 1C19 complex at 37˚C appeared to be more heterogeneous with the 2D classes show different Fab occupancies (S1B Fig). Whereas, the complex at 40˚C showed more homogeneous population (S1C Fig). Only classes that showed high Fab occupancy, indicated by the spikes throughout the surface were selected for further 3D-classification and map reconstruction processes in Relion. Three-dimensional classification was done using the mature DENV2 PVP94/07 28˚C cryo-EM map as a template. Following the 3D refinement in Relion, the orientation search and three-dimensional reconstruction were continued using MPSA and *make3d*. The final maps of DENV2 PVP94/07-Fab 1C19 complex at 37˚C and 40˚C were reconstructed from 8,915 and 15,360 particles, respectively, and the reconstructed maps have a resolution of ~23 Å (S3C and S3D Fig). The resolution was estimated using the same method mentioned earlier.

## Model fitting

To fit the Fab molecule into the corresponding density, a homology model was built by using the Swiss-model server [44]. The homology modeling of Fab 1C19 heavy and light chain

variable regions was based on the crystal structures of two human antibodies (PDB ID 4HIE and 4KQ3), respectively. These two crystal structures were chosen because they have the highest sequence homology to the Fab 1C19 heavy and light chains. Fab was placed into the cryo-EM map manually using the "O" program [45]. The cryo-EM DENV1 structure (PDB ID 4CCT) was used in the interpretation of the cryo-EM density map. Identification of the correct hand of the map was evaluated by fitting the E protein molecule in the map with different handedness. The correct hand was decided by reasonable fitting of E protein that did not involve large movements of E protein, and also when the position of the Fab molecules showed closed proximity to the putative epitope. Using the un-flipped map, the putative epitope (determined by HDXMS studies) on E protein mol A was already located adjacent to the base of Fab density (S8 Fig), but the position of mol C was devoid of density. Slight movement of the position of mols A and C was required to place mol A right in the base of the Fab density and mol C into the positive density (Fig 4B). Mol B and B′ dimers retained their initial positions. Fitting of molecules into the map with opposite handedness (flipped map) showed that large movements of all E proteins in the asymmetric unit were required to fit into the density and this resulted in clashes between the B and B′ molecules, signifying that this configuration is unlikely to occur. This suggested that the un-flipped cryo-EM density map has the correct handedness.

As for the Fab molecule, there were two possible ways to fit the molecule, related by a 180° rotation. The correlation coefficient of the two fits differed only slightly, *i.e.*, 0.911 vs. 0.909 as calculated using "Fit in Map" tool in Chimera [46]. However, surface charge analysis of the interacting surface between E protein and Fab molecule suggested that the first fit had a better charge-complementarity (Fig 6B).

For the Class II structure, the fitting was initiated by placing the mols B-B′ dimer into the densities near to the 2-fold vertices (S4A Fig). The E protein mols A-C′ were then placed at higher radii and fitted into the density. The E protein mol A-Fab 1C19 structure obtained from model fitting of the Class I structure was used as a rigid model in the fitting of E-protein mols A and C′ and the Fab molecules that bound onto them. The fitting of E protein mols A and C′ and the Fab molecules that bound onto them were done in two ways. First it was done by fitting E protein mol A-Fab 1C19 and E protein mol C-Fab 1C19 models into the densities, separately. Using this way gave a correlation coefficient of 0.75 (S1 Table), while some part of the Fab model that attached to mol A was located outside the densities. In the second way, E protein mol C-Fab 1C19 and E protein mol A only were fitted into the density. The second way resulted in a well fitted model with a correlation coefficient of 0.77 (S1 Table) and all densities were interpreted with none of the models or parts of the models is in the negative density. This model was also clash-free with structures from adjacent icosahedral asymmetric unit.

The maps of DENV2 PVP94/07-Fab 1C19 complex at 37°C and 40°C were not fitted with structure models due to the limited available resolution of the maps.

## Plaque reduction neutralization test (PRNT)

Two-fold serial dilutions of HMAb 1C19 were made, and each dilution was mixed with an equal volume of virus and incubated at 37°C for 30 min (final concentration of 1C19 from 10 μg/mL to 4.88 ng/mL). For the neutralization test on DENV2 PVP94/07, additional samples with the complex mixture incubated at 40°C for 30 min were also prepared. BHK-21 cell culture monolayers on 24-well plates were then inoculated with the antibody-virus mixture at 37°C for 1 h. The cell monolayers were washed with PBS to remove unbound virus and overlaid with methylcellulose. Following incubation at 37°C for 3–4 days, plaques were visualized by staining the cell monolayers with crystal violet-formaldehyde. Percentage neutralization

was calculated and triplicate sets of data were used to obtain the $IC_{50}$ value for neutralization for each virus strain.

## Biolayer interferometry

BLI experiments were performed using an Octet Red96 (ForteBio) with anti-human IgG Fc capture coated biosensor tips (AHC biosensors, ForteBio). The 96-well black plates were filled with 200 μL of solution (buffers, rE protein, or IgG 1C19) and agitated at 1000 rpm. For all experiments, anti-human Fc biosensor tips were hydrated in a Tris buffer containing 20 mM Tris pH 8.0, 200 mM NaCl and 0.025% Tween-20 for 1 h at room temperature. The recombinant E proteins and IgG 1C19 were also diluted in the Tris buffer. Dynamic light scattering measurements of the E-proteins showed hydrodynamic size of 74–80 nm. With calculated dimensions of E protein dimer of 140 Å x 50 Å x 38 Å (average dimension of 76 Å, based on the crystal structure of E protein dimer with PDB ID: 1OAN), the DLS results suggested that the aggregation state of the recombinant E proteins is as dimer in the Tris buffer used.

Following a baseline measurement of the biosensor tips in the Tris buffer, the tips were incubated in 10 μg/mL antibody to achieve ~0.6 nm response. Antibody bound tips were then dipped into the Tris buffer to remove excess of IgG 1C19 for 1 min, followed by a 200 s baseline measurement. The association step was done on a two-fold serially diluted rE-proteins at concentrations ranging from 1.563 to 200 nM for 200 s. The tips were then dipped back into the Tris buffer for 1200 s for the dissociation step. Baseline subtraction was performed with tips dipped into buffer in the absence of analyte.

## Glutaraldehyde cross-linking

The oligomerization state of DENV1 PVP159 rE protein at a high concentration (similar concentration used in HDXMS experiment) and a low concentration (similar concentration used in biolayer interferometry experiment) and the ability of HMAb 1C19 to bind to different oligomeric states of the rE protein were investigated through glutaraldehyde cross-linking experiment followed by SDS-PAGE (Coomassie blue staining and Western blot). The initial concentration of rE protein used in HDX-MS experiment was 4.8 mg/mL with final concentration after complexing with Fab 1C19 of 2.4 mg/mL. Meanwhile, the concentrations of rE protein used in biolayer interferometry experiments (Fig 2) were ranging from 0.07 μg/mL to 8.7 μg/mL. DENV1 PVP159 rE-protein was buffer exchanged to crosslinking buffer containing 0.2 M NaHCO$_3$, 0.5 M NaCl, pH 8.3 and concentrated to ~3.3 mg/mL. The cross-linking was done with glutaraldehyde final concentration of 0.1% and rE protein of 9 μg/mL or 3 mg/mL. Five time points were collected i.e., 1, 5, 15, 30 and 60 min and then the reaction was stopped by adding stopping buffer (0.5 M ethanolamine, 0.5 M NaCl, pH 8.3). A sample with the stopping buffer was mixed prior to the addition of gluataraldehyde was prepared as 0 min control. Semi-native non-reducing SDS-PAGE was done. The samples were not boiled and without the addition of any reducing agent. Furthermore, SDS concentration was reduced from 0.1% final concentration to 0.05%, a condition that is close to the native SDS-PAGE method that was introduced by Nowakowski *et al.* [47]. Non-reducing and low SDS concentration condition would preserve or partially preserve the rE protein structure during the run thus, for western blot, HMAb 1C19 is still able to bind to rE protein. Samples were mixed with 5X Pierce lane marker non-reducing sample buffer (Thermofisher) to a final concentration of 0.5X and glycerol was further added to a final concentration of 5%. Samples were loaded into 4–20% precast polyacrylamide gel (Biorad). For cross-linked rE protein at concentration of 3 mg/mL, two gels were prepared, one was stained by Coomassie blue (InstantBlue, Expedeon), whereas the other was used for Western blot analysis. For cross-linked rE protein at concentration of 9 μg/

mL, only western blot analysis was done. Western blot analysis was done using HMAb 1C19 as primary antibody at concentration of 2 μg/mL and goat anti-human IgG (H+L) HRPO conjugate (Invitrogen) as secondary antibody at ratio of 1:2000.

## Atomic-resolution MD simulations

The monomeric DENV1 E protein ectodomain (residues 1–394) coordinates were obtained from the cryo-EM structure. The Rosetta software suite of programs was used to refine the raw fit of the E-Fab 1C19 complex to the cryo-EM map. Rosetta was configured for local docking with 500 conformations generated. Monte Carlo position sampling parameters were set to a maximum 3 Å for translation and 8˚ for rotation per move. All 500 conformations were scored using the Rosetta energy function and ranked by interaction energy and Cα RMSD. The system resulted in a complex between the DENV1 E protein dimer and a single Fab 1C19. In order to obtain the DENV2 NGC strain of E protein complexed with Fab 1C19, we performed *in silico* mutations on the DENV1 E protein dimer. Each complex was placed in the center of a cubic box (170 x 100 x 120 Å$^3$) and then solvated with approximately 60,000 TIP3P [48] water molecules. All simulations were performed using the Amber99SB*-ILDN-Q forcefield [49]. Ionizable residues together with the termini were treated in their fully charged state according to neutral pH. Sodium and chloride ions were added to reach 0.1 M salt concentration whilst neutralizing the overall system charge. MD simulations were performed with the GROMACS 2016 package [50]. Equations of motion were integrated through the Verlet leapfrog algorithm with a 2 fs time step. Bond lengths involving hydrogens were constrained with the LINCS algorithm [51]. The cutoff distance was switched from 9 to 12 Å for the short-range neighbor list and for van der Waals interactions. The Particle Mesh Ewald (PME) [52] method was applied for the long-range electrostatic interactions with a 12 Å real space cutoff. The velocity rescale thermostat with an additional stochastic term [53] and Parinello-Rahman [54] barostat were used to maintain the temperature at 310 K and pressure at 1 bar, respectively. Initial velocities were set according to the Maxwell distribution. Periodic boundaries were applied in all directions. Initial configurations were minimized using the steepest descent algorithm followed by equilibration with position restraints on protein heavy atoms with a force constant of 1,000 kJ mol$^{-1}$ nm$^{-2}$ in the *NVT* and subsequently *NPT* ensembles, for 1 ns and 5 ns, respectively. The production run was set to 1.1 μs in the *NPT* ensemble with the following strategy: i) 0.55 μs with weak position restraints applied to the E protein (force constant of 100 kJ mol$^{-1}$ nm$^{-2}$), while the Fab 1C19 was free to move without restraints; and ii) 0.55 μs with weak position restraints applied to the E protein domain I and II Cα atoms (force constant of 100 kJ mol$^{-1}$ nm$^{-2}$), while the 1C19 Fab and E protein domain III was free to move without restraints. The trajectory was saved every 0.005 μs resulting in a total of 2,200 frames. All simulations were performed on the National Supercomputing Centre (https://www.nscc.sg) Linux cluster. Each simulation employed 4 nodes consisting of 1 GPUs (Nvidia Tesla K40t) and 24 CPUs (Intel Xeon CPU E5-2690 v3 @ 2.6 GHz) each.

## MD simulation analysis

Per-residue E protein contacts with the Fab 1C19 (epitope) were calculated based on a 0.6 nm cut-off distance. Any E protein residue which was at least once within this cut-off distance of the Fab 1C19 during any frame across the entire combined trajectory of 1.1 μs was considered as a possible epitope. All structural analyses were performed using tools within the GROMACS and VMD packages. The angle between the E plane vector and the Fab vector was calculated according to the following assumptions: (i) the E protein plane was defined as an average of the center of mass of domain II (chain A)–domain III (chain A)–domain II (chain B) and

domain II (chain B)–domain III (chain B)–domain II (chain A); (ii) the vector for the Fab long-axis was defined from the center of mass of residues 1–105 of either antibody chain (interacting part), to the center of mass of residue 120 to the C-terminus of either antibody chain (upper part). The extremes of the trajectory of the two most dominant motions for protein backbone atoms were extracted using principal component analysis [55] over the entire, combined 1.1 μs trajectory.

## Supporting information

**S1 Fig. The 2D class average particles of Fab 1C19 complexed with DENV1 strain WestPac 74 at 37˚C and DENV2 strain PVP94/07-Fab 1C19 at 37˚C or 40˚C. (A)** The 2D class averages of DENV1 strain WestPac 74-Fab 1C19 complex particles at 37˚C. All classes showed particles with likely high Fab 1C19 occupancy (red box) as indicated by the rough virus surfaces. The number of particles in each 2D classes is indicated. (**B**) The 2D class averages of DENV2 PVP94/07-Fab 1C19 complex particles at 37˚C. The images showed heterogeneous populations of the complex particles at various occupancies—from uncomplexed (green) to partial (yellow box) or high occupancies. (**C**) The 2D class averages of DENV2 strain PVP94/07-Fab 1C19 complex particles at 40˚C. Most classes showed particles with bound Fab 1C19 at high occupancy, and only a few unbound virus particles were left in the population.
(TIF)

**S2 Fig. Analysis of the oligomerization states of DENV1 PVP159 rE protein at different concentrations by glutaraldehyde cross-linking experiments. (A)** Cross-linked DENV1 PVP159 rE protein concentration at 3 mg/mL similar to the that used in HDXMS experiments. Visualization of the E protein bands on non-reducing SDS-PAGE gel by either Western blot using HMAb 1C19 (left) or by staining with Commasie blue (right). Results show the presence of monomeric, dimeric and higher oligomeric states of E proteins in the protein sample at 3mg/mL and HMAb 1C19 is able to detect these different oligomerization states. (**B**) Similar crosslinking experiment is also done with DENV1 rE protein concentration at 9 $\mu$g/mL similar to that used in BLI experiments. At this concentration, E protein exists as monomeric state. The SDS-PAGE experiments were done in non-reducing condition (not boiled and without reducing agent eg. DTT or $\beta$-mercaptoethanol).
(TIF)

**S3 Fig. Resolution of the cryoEM maps.** Fourier Shell Correlation (FSC) curve of: (**A**) Class I map of DENV1 strain WestPac 74-Fab 1C19 complex particles at 37˚C. (**B**) Class II map of DENV1 strain WestPac 74-Fab 1C19 complex particles at 37˚C. (**C**) Map of DENV2 strain PVP94/07-Fab 1C19 complex particles at 37˚C. (**D**) Map of DENV2 strain PVP94/07-Fab 1C19 complex particles at 40˚C. The FSC curve was plotted from two reconstructed maps of two half-datasets of the final iteration step and the resolution was estimated by using a 0.5 cut-off value.
(TIF)

**S4 Fig. Fitting of the model into Class II DENV1 strain WestPac 74-Fab 1C19 at 37˚C cryoEM map. (A)** The initial fitting was done by fitting the E protein mols B-B′ dimer into the density located on the 2-fold vertices and followed by fitting the E protein mols A-C′ dimer into the density located at a higher radius. The protruding density that belongs to the Fab 1C19 density, which is not occupied by the E protein mols A-C′ dimer, is indicated by a magenta arrow. (**B**) The E protein from the E protein mol A-Fab 1C19 model of Class I complex structure was superimposed onto both mols A and C′ E proteins, and then each of the E protein molecule-Fab 1C19 complex models was fitted as a rigid body separately into the

density. Part of Fab 1C19 binding to E protein mol A is outside the density (indicated by orange arrows), suggesting that the Fab may not be present. (**C**) The final fit of the E proteins with Fab bound to only mol C into the density map is shown.
(TIF)

**S5 Fig. MD simulations of DENV1 and DENV2 NGC strain interacting with Fab 1C19. (A)** The angle between the E protein dimer plane and the Fab long-axis vector over the simulation time for DENV1. (**B**) Root-mean-square-deviation (RMSD) of Fab backbone atoms on DENV1 and DENV2 dimeric E protein; this indicates that the DENV2 complex is less stable. (**C**) Epitope of 1C19 Fab on DENV2 E protein dimer. The top view protein is shown in cartoon representation (domain I: red, domain II: yellow, domain III: blue) with distinctive Fab interacting regions (shown as spheres).
(TIF)

**S6 Fig. Amino acid sequence alignment between E proteins ectodomain of DENV1 PVP159 and WestPac 74 strains, DENV2 PVP94/07 and NGC strains, DENV3 863DK strain and DENV4 2270DK strain.** The amino acid residues on *b* strand, *bc* loop-*c* strand-*cd* loop and *d* strands, and *hi* and *ij* loops regions, which are estimated to be the location of the epitope of HMAb 1C19, are indicated in a box outlined with dashed green. Conserved amino acids are shown as letters in white font with red background, whereas partially conserved amino acids are shown as letters in red font with white background. This sequence alignment shows that the amino acid residues in the epitope region are highly conserved for different strains within the same serotype, but there are some differences across the two serotypes that may lead to different binding affinity to HMAb 1C19.
(TIF)

**S7 Fig. Superposition of the sub-nanometer resolution cryo-EM structure of un-complexed mature DENV1 E protein (pre-fusion structure) (PDB ID: 4CCT) onto the crystal structure of post-fusion trimeric rE protein of DENV1 (PDB ID: 4GSX).** The superposition showed that the trimeric post-fusion E protein (one of which is colored in cyan, while the other molecule in grey) has a vastly different *ij*-loop conformation compared to the pre-fusion E protein (orange). The *hi*-loop also showed a different conformation. These features are both part of the 1C19 epitope identified by HDXMS. Both *hi* and *ij* loops are indicated.
(TIF)

**S8 Fig. Stereoview of the initial fitting of E protein molecules into the reconstructed cryo-EM map of Class I particles.** The unexpanded mature DENV1 structure (PDB ID 4CCT) in the Class I Fab 1C19:DENV1 WestPac complex map before fitting. The location of the putative epitope (green spheres) on all E protein molecules identified by HDXMS. Peptide 219–243 (green spheres), which was identified as contributing to the epitope by HDXMS studies, is located closest to the base of the Fab densities (purple arrow). Therefore the antibody most likely binds to mol A. However, some slight translation of the mol A was required in order to place the epitope at the foot of the antibody. Part of mol C is located in negative densities (black arrows) and therefore, some adjustments of position were required. Three individual E protein molecules in an asymmetric unit are labelled as A, B and C, whereas the corresponding molecules in a neighboring asymmetric unit within a raft are labelled as A′, B′ and C′. DI, DII or DIII of the E protein are colored in red, yellow or blue, respectively.
(TIF)

**S1 Table. Correlation coefficient of the fitted models to the density maps.**
(PDF)

**S1 Movie. The filtered trajectory extremes of the most dominant motion (representing ~40% of the variance across the entire simulation).** Protein is shown in cartoon representation: orange and grey for two DENV1 E protein monomers, green for Fab 1C19.
(MOV)

**S2 Movie. The filtered trajectory extremes of the second most dominant motion (representing ~35% of the variance across the entire simulation).** Protein is shown in cartoon representation: orange and grey for two DENV1 E protein monomers, green for Fab 1C19.
(MOV)

## Acknowledgments

DENV1 strain PVP159 and DENV2 strain PVP94/07 were kindly provided by Eng Eong Ooi (Duke-NUS, Singapore). P.J.B, R.G.H and J.K.M would like to acknowledge National Supercomputing Centre (https://www.nscc.sg) for providing computational resources. We acknowledge Eva Harris for the original donor from her cohort studies and also Aravinda de Silva for his role in the work related to the original characterization of the 1C19 antibody.

## Author Contributions

**Conceptualization:** James E. Crowe, Jr., Shee-Mei Lok.

**Data curation:** Guntur Fibriansah, Elisa X. Y. Lim, Jan K. Marzinek, Thiam-Seng Ng, Joanne L. Tan, Roland G. Huber, Xin-Ni Lim, Victor A. Kostyuchenko.

**Formal analysis:** Guntur Fibriansah, Elisa X. Y. Lim, Jan K. Marzinek, Joanne L. Tan, Roland G. Huber, Xin-Ni Lim, Victor A. Kostyuchenko, Ganesh S. Anand, Peter J. Bond, James E. Crowe, Jr., Shee-Mei Lok.

**Funding acquisition:** Ganesh S. Anand, Peter J. Bond, James E. Crowe, Jr., Shee-Mei Lok.

**Investigation:** Guntur Fibriansah, Elisa X. Y. Lim, Jan K. Marzinek, Thiam-Seng Ng, Joanne L. Tan, Roland G. Huber, Xin-Ni Lim, Victor A. Kostyuchenko, Peter J. Bond, James E. Crowe, Jr., Shee-Mei Lok.

**Methodology:** Guntur Fibriansah, Elisa X. Y. Lim, Jan K. Marzinek, Thiam-Seng Ng, Joanne L. Tan, Roland G. Huber, Xin-Ni Lim, Valerie S. Y. Chew, Victor A. Kostyuchenko.

**Project administration:** James E. Crowe, Jr., Shee-Mei Lok.

**Resources:** Jian Shi, Ganesh S. Anand, Peter J. Bond, James E. Crowe, Jr., Shee-Mei Lok.

**Supervision:** Peter J. Bond, James E. Crowe, Jr., Shee-Mei Lok.

**Validation:** Guntur Fibriansah, Jan K. Marzinek, Roland G. Huber, Ganesh S. Anand, Peter J. Bond, Shee-Mei Lok.

**Visualization:** Guntur Fibriansah, Elisa X. Y. Lim, Jan K. Marzinek, Thiam-Seng Ng, Shee-Mei Lok.

**Writing – original draft:** Guntur Fibriansah, Jan K. Marzinek, Peter J. Bond, Shee-Mei Lok.

**Writing – review & editing:** Guntur Fibriansah, Jan K. Marzinek, Peter J. Bond, James E. Crowe, Jr., Shee-Mei Lok.

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
