## [Decision Letter · Decision Letter 0]

19 Nov 2020

Dear Dr. Lok,

Thank you very much for submitting your manuscript "Antibody affinity versus dengue morphology influences neutralization" for consideration at PLOS Pathogens. As with all papers reviewed by the journal, your manuscript was reviewed by members of the editorial board and by several independent reviewers. The reviewers appreciated the attention to an important topic. Based on the reviews, we are likely to accept this manuscript for publication, providing that you modify the manuscript according to the review recommendations.

The manuscript was reviewed by three individuals with expertise in the structure and biology of dengue virus and the challenges faced in developing dengue virus vaccines. Overall, all the three reviewers were quite positive in their assessment of the work, commenting favorably on the nature of the experimental approach and the significance of the findings. However, reviewers did raise issues in a few areas that need to be addressed. For example, (i) Reviewer 2 was not convinced of the inference brought up in the Discussion that the 1C19 antibody could serve as an effective therapeutic, given the results of previous studies suggesting the opposite, (ii) Reviewer 1 wanted additional information regarding the interpretation of the cryoEM work, and (iii) Reviewer 1 also thought key details were missing for the BL1 and HDMXS experiments that are necessary for analyzing the their results. Although these are generally minor issues, they will need to be thoroughly addressed and the manuscript appropriately modified before further consideration for publication. Thank you again for submitting such an outstanding manuscript to the journal. We look forward to receiving the revised manuscript in the near future and its ultimate publication.

Sincerely,

John T. Patton, PhD

Associate Editor

PLOS Pathogens

Mark Heise

Section Editor

PLOS Pathogens

Kasturi Haldar

Editor-in-Chief

PLOS Pathogens

orcid.org/0000-0001-5065-158X

Michael Malim

Editor-in-Chief

PLOS Pathogens

orcid.org/0000-0002-7699-2064

Thank you very much for submitting your manuscript "Antibody affinity versus dengue morphology influences neutralization" (PPATHOGENS-D-20-02220) for consideration by PLOS Pathogens. The manuscript was reviewed by three individuals with expertise in the structure and biology of dengue virus and the challenges faced in developing dengue virus vaccines. Overall, all the three reviewers were quite positive in their assessment of the work, commenting favorably on the nature of the experimental approach and the significance of the findings. However, reviewers did raise issues in a few areas that need to be addressed. For example, (i) Reviewer 2 was not convinced of the inference brought up in the Discussion that the 1C19 antibody could serve as an effective therapeutic, given the results of previous studies suggesting the opposite, (ii) Reviewer 1 wanted additional information regarding the interpretation of the cryoEM work, and (iii) Reviewer 1 also thought key details were missing for the BL1 and HDMXS experiments that are necessary for analyzing the their results. Although these are generally minor issues, they will need to be thoroughly addressed and the manuscript appropriately modified before further consideration for publication. Thank you again for submitting such an outstanding manuscript to the journal. We look forward to receiving the revised manuscript in the near future and its ultimate publication.

Reviewer Comments (if any, and for reference):

Reviewer's Responses to Questions

**Part I - Summary**

Reviewer #1: Fibriansah et al. use a series of complementary approaches to determine where the antibody 1C19 binds to the dengue E protein. They use two strains of DENV1 and two strains of DENV2 for their studies since these viruses bind to the antibody differently at 37C. The authors compare and contrast the results from each method. The work provides a basis for designing antibodies that can be used as therapeutics during dengue infection.

Reviewer #2: Human MAb 1C19 was isolated previously from a secondary dengue patient. This antibody binds to an epitope that is conserved between serotypes. Moreover, the antibody cross neutralizes serotypes 1-3 in cell culture. In this manuscript, Fibriansah et al map the binding site of 1C19 on dengue type 1 and 2 using deuterium exchange spectroscopy and cryo EM. The results demonstrate that the antibody binds a complex cryptic epitope on domain II of E protein. The exposure of the epitope was dependent on temperature, the morphology of the virus and the serotype. The strengths of the paper are the excellent and complementary approaches used to map the epitope and the design of studies to link viral envelope dynamics to antibody binding.

The main weakness of the paper is a superficial discussion of how this antibody might be used as a therapeutic antibody. The studies reported here as well as previous work with this antibody strongly argue against any useful role for this (and related antibodies) in protection as discussed below.

Reviewer #3: This work characterizes the neutralizing human mAb 1C19 for four strains belonging to two dengue serotypes, DENV1 and DENV2. The authors mapped the 1C19 epitope to the DENV E protein domain II using hydrogen-deuterium exchange mass spectrometry, cryoEM, and molecular dynamics. The 1C19 binds to DENV1 E protein with higher affinity compared to that of DENV2, and displays strong neutralizing activities for all DENV1 strains regardless of their morphologies, suggesting the tighter binding of 1C19 can outcompete the E-E interactions. This study is well-executed and provides structural and mechanistic insights into the neutralization of DENV by a human mAb.

**Part II – Major Issues: Key Experiments Required for Acceptance**

Reviewer #1: For the two classes of structures determined by cryoEM, are the authors proposing these occur individually? Or Class II is an intermediate on the way to achieving Class I? Or vice versa, Class I occurs and induces the change seen in Class II?

The authors conclude antibody binding to E is a dynamic process and requires several discrete conformational changes by both the antibody and the E protein. Can the authors propose a model of how the particles move or undergo structural transitions based on their data? For example, using the data from DENV1 can the authors propose how 1C19 binds to DII, when it might interact and then dissociate with DIII, how E moves and how that may neutralize infection? The authors address parts of this question throughout the manuscript, I think it would beneficial if the authors could propose 1 (or 2 or 3) models of what might be happening.

When fitting the E protein into the cryoEM maps of DENV1, did the E protein monomer have to be rotated at all, thus exposing different residues compared to its conformation when a dimer?

Was the rE protein used for BLI and HDXMS experiments a monomer or a dimer or a mixture? If the E was a monomer, there are fewer steric constraints than if a dimer and higher likelihood for Fab to bind to the protein. In contrast, when an Fab binds to an E protein in a virus, a slight conformational change in one protein may have a more profound effect on neighboring E proteins because of the close packing. This was evident in the cryo structures. I also ask about this because HDXMS showed residues 219-243 as being the main Fab footprint. However, the cryo structures don’t directly show this footprint (near) and the MD simulations identify 239-252, slightly different.

Have the authors mutated the residues from DENV4 into DENV1 to see if the slight changes in surface residues is enough to reduce antibody binding?

Reviewer #2: While no additional experiments are required, I do not agree with the conclusion in the abstract and discussion that these studies support the use of this antibody as a therapeutic. The dengue field is currently contending with the fact that in vitro antibody neutralization is not a reliable predictor of protection in vivo. Indeed, this study highlights how dynamic properties of lab strains of dengue viruses can lead to neutralization in cell culture by mechanisms that have no relevance to protection in vivo.

1) In the original publication describing 1C19 (Smith et al), the antibody given prophylactically to mice had a modest effect on DENV1 (1 log reduction) and DENV2 (2.5 log reduction) viremia. This effect is mild compared to other human antibodies that provide close to sterilizing immunity when administered prophylactically.

2) Tuan et al compared a large panel of human MAbs for their ability to directly neutralize DENVs in blood collected from viremic dengue patients. In that study many antibodies such as 2D22, 14C10, EDE Mabs neutralized "in vivo" virus. 1C19 had no effect on any DENV serotype ( Tuan Vu T et al. (2019) Blockade of dengue virus transmission from viremic blood to Aedes aegypti mosquitoes using human

monoclonal antibodies. PLoS Negl Trop Dis 13(11): e0007142).

The investigators need to frame their work against this body of work as well as recent vaccine trials demonstrating cell culture neutralization is not always predictive of protection in vivo.

Reviewer #3: (No Response)

**Part III – Minor Issues: Editorial and Data Presentation Modifications**

Reviewer #1: The concluding two paragraphs of the discussion really emphasize the rationale of this work and the potential therapeutic benefits. I would like to see that brought up in the introduction. Maybe some mention of dengue treatments and why antibody treatment is worth pursuing.

Reviewer #2: (No Response)

Reviewer #3: Figure 1:The authors should include scale bars in all cryoEM micrographs.

Table 1 may also include the Kd value of 1C19 to E proteins to show its correlation with the IC50 of each strain.

PLOS authors have the option to publish the peer review history of their article (what does this mean?). If published, this will include your full peer review and any attached files.

Reviewer #1: No

Reviewer #2: No

Reviewer #3: No
---

## [Editor Report · Decision Letter 1]

25 Jan 2021

Dear Dr. Lok,

We are pleased to inform you that your manuscript 'Antibody affinity versus dengue morphology influences neutralization' has been provisionally accepted for publication in PLOS Pathogens.

Best regards,

John T. Patton, PhD

Associate Editor

PLOS Pathogens

Mark Heise

Section Editor

PLOS Pathogens

Kasturi Haldar

Editor-in-Chief

PLOS Pathogens

orcid.org/0000-0001-5065-158X

Michael Malim

Editor-in-Chief

PLOS Pathogens

orcid.org/0000-0002-7699-2064

Thank you very much for submitting your revised manuscript entitled "Antibody affinity versus dengue morphology influences neutralization" (PPATHOGENS-D-20-02220R1) for review by PLOS Pathogen. The outstanding efforts made by the authors in responding to the reviewer's comments and concerns are much appreciated. I am pleased to accept the manuscript for publication.
---

## [Editor Report · Acceptance letter]

17 Feb 2021

Dear Dr. Lok,

We are delighted to inform you that your manuscript, "Antibody affinity versus dengue morphology influences neutralization," has been formally accepted for publication in PLOS Pathogens.

Best regards,

Kasturi Haldar

Editor-in-Chief

PLOS Pathogens

orcid.org/0000-0001-5065-158X

Michael Malim

Editor-in-Chief

PLOS Pathogens

orcid.org/0000-0002-7699-2064